# Multiscale networks in Alzheimer's disease identify brain hypometabolism as central across biological scales

**Elena Lara-Simon**[ID][1], **Juan Domingo Gispert**[2], **Jordi Garcia-Ojalvo**[ID][1], **Pablo Villoslada**[ID][1,3]*, **for the Alzheimer's Disease Neuroimaging Initiative**

**1** Department of Medicine and Life Sciences, Universitat Pompeu Fabra, Barcelona, Spain, **2** Barcelona Beta Research Center, Barcelona, Spain, **3** Hospital del Mar Research Institute, Barcelona, Spain

¶ Data used in preparation of this article were obtained from the Alzheimer's Disease Neuroimaging Initiative (ADNI) database (adni.loni.usc.edu). As such, the investigators within the ADNI contributed to the design and implementation of ADNI and/or provided data but did not participate in analysis or writing of this report. A complete listing of ADNI investigators can be found at:
http://adni.loni.usc.edu/wp-content/uploads/how_to_apply/ADNI_Acknowledgement_List.pdf
* pablo.villoslada@upf.edu

**Data availability statement:** The data used in this article was obtained from the Alzheimer's Disease Neuroimaging Initiative (ADNI)

## Abstract

Alzheimer's disease encompasses multiple biological scales, spanning molecular factors, cells, tissues, and behavioral manifestations. The interplay among these scales in shaping the clinical phenotype is not yet fully comprehended. In particular, there is great interest in understanding the heterogeneity of the clinical aspects of AD in order to improve treatment and prevention, by targeting those aspects most susceptible to the disease. Here we employed a systems biology approach to address this issue, utilizing multilayer network analysis and deep phenotyping. This integrative analysis incorporated genomics, cerebrospinal fluid biomarkers, tau and amyloid beta (A$\beta$) PET imaging, brain MRI data, risk factors, and clinical information (cognitive tests scores, Clinical Dementia Rating and clinical diagnosis) obtained through the ADNI collaboration. Multilayer networks were built based on mutual information between the elements of each layer and between layers. Boolean simulations allowed us to identify paths that transmit dynamic information across layers. The most prominent path for predicting variables in the cognitive phenotype layer included the PET radiotracer fluorodeoxyglucose (FDG) in the posterior cingulate. Combinations of different symptomatic variables, mainly related to mental health (depression, mood swings, drowsiness) and vascular features (hypertension, cardiovascular history), were also part of the paths explaining the average phenotype. Our results show that integrating the flow of information across biological scales reveals relevant paths for AD, which can be subsequently explored as potential biomarkers or therapeutic targets. In particular, our results point for paths related with brain hypometabolism as a key feature in AD.

database. Access to ADNI data should be requested at its website: https://adni.loni.usc.edu. Web interfaces of the individual networks, the combined six-layer network, the paths and the top risk factors paths are available at GitHub. The complete code used for the analyses and simulations described in this study is publicly available at the following repository: https://github.com/dsb-lab/multilayerAD. Instructions and a link to the ADNI data access portal are provided in the README of the GitHub repository, so that researchers can request access and reproduce the analyses with the same code.

**Funding:** This work was supported to JGO by project PID2021-127311NB-I00 financed by the Spanish Ministry of Science and Innovation, the Spanish State Research Agency and FEDER (MICIN/AEI/10.13039/ 501100011033/FEDER), by the Maria de Maeztu Programme for Units of Excellence in R&D (project CEX2018-000792-M), and by the ICREA Academia programme. The funders had no role in study design, data collection and analysis, decision to publish, or preparation of the manuscript.

**Competing interests:** I have read the journal's policy and the authors of this manuscript have the following competing interests: PV has received consultancy fees and holds stocks in Bionure Investment, Accure Therapeutics, Attune Neurosciences, QMENTA, CLight, NeuroPrex, Spiral Therapeutics, and Adhera Health, none related to this study. PV holds patent rights and has received royalties and consultancy fees from Oculis Holding AG for using OCS-05 (aka BN201) to treat optic neuritis (NCT04762017). JDG is currently an employee at Astra-Zeneca, but his contribution were done before joining the company. All the other authors have declared that no competing interests exist.

## Author summary

Complex diseases such as Alzheimer's Disease (AD) involve a diverse array of biological processes. In our investigation, we undertook a systems biology approach to AD using network analysis and deep phenotyping within a prospective cohort of patients, incorporating clinical, imaging, genetics, and omics assessments. The gene, molecular and imaging paths explained variation in central nervous system damage, and in metrics of disease severity, pointing to a significant role of energy deficit within brain networks in the development of AD. The elucidation of multilayer paths in this context provides insights into the diverse phenotypes of the disease and holds the potential to improve understanding of its pathogenesis.

## Introduction

Alzheimer's Disease (AD) is a progressive, degenerative brain disease characterized by loss of function and death of nerve cells. The disease is defined by the presence of amyloid beta (A$\beta$) plaques and neurofibrillary tau tangles in the brain [1]. Abnormal deposits of these two proteins have been seen to form aggregates and inclusions, de-structuring the brain architecture. AD is the most common form of dementia, accounting for 60-80% of all cases. Accurate diagnosis is possible in vivo using biomarkers [2–4]. Although early molecular markers exist, even in plasma [5], there remains a strong interest in understanding heterogeneity at all levels in the clinical manifestations of AD, both for treatment and prevention, to identify those individuals at highest risk for the disease. To that end, a multimodal approach integrating different omics data types (genomics, proteomics and metabolomics) and imaging, appears especially useful [6]. Here we follow this approach using multilayer network analysis to represent the flow of events underlying the phenotype of AD, including gene expression, tissue damage, and clinical symptoms. The goal is to identify multimodal paths associated with specific features of AD that will help explain the observed clinical heterogeneity of the disease, and identify candidate paths for personalized interventions.

Modern complex network theory has shown to be a very useful tool for comprehending the intricate architecture of biological processes. It is not sufficient to identify and classify the system's constituent parts to fully comprehend complex biological systems; understanding the interactions between those elements is also necessary [7,8]. However, it can be challenging to evaluate the interaction patterns and functional architecture of biological systems due to the nontrivial nature of those interactions, the limited statistical power of clinical data, and the inherent nonlinearities in the dynamics of individual elements.

In spite of those difficulties, some studies have already shown the usefulness of multilayer networks to help improve our understanding of disease pathogenesis. Multilayer networks have been used, for instance, to directly link the genomic layer with the phenotypes in different types of cancer [9]. In a different approach, a topological analysis of a bipartite network linking drugs and proteins showed an overabundance of drugs targeting the same proteins, suggesting more functional drugs for more diverse targets [10]. Multilayer network analysis has also been applied to a cohort of multiple sclerosis patients, enabling the identification of genetic, protein, and cellular paths explaining variation in central nervous system damage and disease severity metrics, and highlighting their potential as biomarkers [11]. Specifically in Alzheimer's disease, a multidimensional network framework enabled detection of the disease with 90% accuracy, revealing unique insights into disease heterogeneity through identification of similar subtypes with diverse biomarker profiles [12].

In order to better understand the relationship between endotype and phenotype, our strategy focuses on connecting the microscopic and macroscopic dimensions of AD. Our approach enables us to integrate all available biological scales through a multilayer network that allows a multi-domain analysis of a wide variety of AD features. This form of analysis is useful for identifying how the disease phenotype is manifested at different levels, so that new treatment horizons may be explored and approached. Our project is hypothesis-based, aiming at using real-world data to evaluate our hypothesis and uncover a plausible explanation.

Our approach is divided in the following steps. First, we construct networks for each of the biological layers individually, using mutual information as the criterion to connect elements of each layer. Next we identify the connections between layers in order to obtain a multilayer network. An unbiased analysis of the network connectivity reveals a modular structure that supports the hypothesis that a hierarchy exists among the different biological layers. Finally, we identify paths and key drivers for AD phenotype expression. To that end, we use dynamical Boolean simulations to calculate the shortest paths of information transmission that lead to the phenotype layer. The variables most commonly present in those paths can be considered predictors of the phenotype, shedding light on how the different scales interact to produce this complex disease, and potentially enabling its diagnosis.

## Materials and methods

### Patients

Data used in the preparation of this article were obtained from the Alzheimer's Disease Neuroimaging Initiative (ADNI) database https://adni.loni.usc.edu/. The ADNI was launched in 2003 as a public-private partnership, led by Principal Investigator Michael W. Weiner, MD. The primary goal of ADNI has been to test whether serial magnetic resonance imaging (MRI), positron emission tomography (PET), other biological markers, and clinical and neuropsychological assessment can be combined to measure the progression of mild cognitive impairment (MCI) and early Alzheimer's disease (AD) [13]. Specifically, we used data on patients between the ages of 55 and 90 for over a decade at 57 sites in the US and Canada. The cohort included 1998 participants, of which 807 were defined clinically as MCI patients, 533 were AD patients, and 622 were controls. The features used to implement the multilayer network included: MRI (Magnetic Resonance Imaging), PET (Positron emission tomography) scans, CSF (cerebrospinal fluid)/plasma proteomics, genetic information, risk factors, cognitive tests and diagnosis assessment.

**Genetic layer.** This layer contains genetic information such as *APOE* and *TOMM*40 alleles [14], Polygenic Hazard Score (PHS) and Cumulative Incidence Rate (CIR). The genetic network, as well as the rest of the individual networks, can be seen in Fig 6, where nodes represent variables and the edges represent the mutual information between pairs of variables.

The dataset contains three variables in particular that refer to the *APOE* gene: the individual copies *APOE_A*1 and *APOE_A*2, for which the number refers to the type of allele that forms it, and *APOE*, that represents the amount of E4 alleles present in the individual, which can be 0, 1 or 2. For example, if a subject has *APOE_A*1 = 2 (has allele E2) and *APOE_A*2 = 4 (has allele E4), then *APOE* = 1 (one E4 allele), indicating the individual is heterozygous for the E4 allele.

The polygenic hazard score is a measure based on a combination of multiple genetic variants, developed to quantify the age of onset of AD dementia. Several SNPs were examined for their association with AD, then a stepwise Cox proportional hazard model was applied to choose the SNPs that improved the model. Finally, the vector product between the genotype for the SNPs and the AD-incidence rates provides quantitative estimates of the annualized

(cumulative) incidence rate [15]. For this reason, we decided to also add the Cumulative Incidence Rate (CIR) to the genetic layer, even though it could be considered a risk factor, because it is closely related to PHS. Importantly, PHS was associated with in vivo biomarkers of AD pathology such as reduced CSF $A\beta42$ and elevated CSF total tau across the AD spectrum (in older controls and dementia individuals) [16].

**Molecular layer.** For both $A\beta$ and tau, CSF as well as plasma samples where used in order to extract the subjects' measures. For $A\beta$, two types of measures where used: $A\beta42$ and the ratio $A\beta42/A\beta40$. As for tau, both tau and phosphorylated tau are measured. Other molecular measures are amyloid precursor protein (APP); $\beta$-secretase, an enzyme that participates in the APP to $A\beta42$ pathway, and even though it is part of the genetic information of the cells, telomere length (TL) is also considered a molecular biomarker, since it does not really act as a gene but a protection of the genetic material, thus a protection of the cells functions and regulations. There is also a variable that represents the ratio between TL and the Single Copy Gene ratio (QPCR), another measure for TL.

**PET layer.** Acquisition and standardized preprocessing steps of MRI and PET data in ADNI have been reported previously and are described in detail on the ADNI website [17]. Only the variables that have the largest relevance in the development of the disease are considered. Braak stages were used for tau [18], and the Landau signature was used for FDG, which included the following regions: right and left angular gyrus, bilateral posterior cingulate, and right and left inferior temporal gyrus [19]. Finally, regions used for $A\beta$ were: anterior cingulate cortex, isthmus cingulate cortex, posterior cingulate cortex, inferior frontal gyrus (pars opercularis, pars triangularis, and pars orbitalis), lateral orbitofrontal cortex, medial orbitofrontal cortex, middle frontal gyrus (caudal and rostral middle frontal), superior frontal gyrus, frontal pole, inferior temporal gyrus, middle temporal gyrus, superior temporal gyrus (superior temporal and transverse temporal), fusiform gyrus, entorhinal cortex, parahippocampal gyrus, lingual gyrus, lateral occipital gyrus, temporal pole, insula, inferior parietal gyrus, supramarginal gyrus, precuneus, superior parietal gyrus, precentral gyrus, postcentral gyrus, paracentral gyrus and cuneus [17].

**MRI layer.** Cortical thicknesses from two signature AD regions were taken into account. First, Dickerson's signature, which refers to brain regions known to be highly atrophied due to AD, making them susceptible to thinning in subjects who might be in very early stages. Therefore, tracking the thickness of these regions years before symptoms appear could help in early detection and intervention. It encompasses all parietal as well as frontal regions and the supramarginal [20]. The second one, Jack signature, involves the entorhinal, inferior temporal, middle temporal, inferior parietal, fusiform and precuneus areas [21]. Cortical volume, thickness average, and standard deviation were used for all those regions, and also total brain gray and white matter volume as well as CSF volumes. Lastly, we also collected data from tensor based morphometry and atrophy measures.

**Phenotype layer.** This is considered to be the top layer of our multilayer model. On the one hand, we have all the cognitive tests scores: Alzheimer Disease Assessment Scale (ADAS) [22], Mini-Mental State Examination (MMSE) [23], Montreal Cognitive Assessment (MOCA), the composite executive function score, composite language, memory and visuospatial scores. On the other hand, there are also two diagnosis variables, which indicate the level of dementia of the subject: the Clinical Dementia Rating (CDR) and the clinical diagnosis [24]. It defines the group the subject belongs to: controls, Mild Cognitive Impairment (MCI) patients or AD patients.

**Risk factors layer.** This layer contains a variety of different aspects such as comorbidities, clinical history, demographics, depression, and more. The information in the majority

of these variables is either binary (with a value of 0 or 1 representing if there is an absence or existence of a given risk factor, respectively), or categorical, like gender (*PTGENDER*), handedness (*PTHAND*) and marital status (*PTMARRY*). Additionally, we also have some quantitative measures such as birth year (*PTDOBYY*), years of education (*PTEDUCAT*), and features given by scores, such as the total modified Hachinski score, which represents a sum of all modified Hachinski comorbidities. Hachinski scores are a clinical tool to differentiate types of dementia. In particular, modified Hachinski scores differentiate Alzheimer's type dementia and other dementias [25].

## Data processing

The omics, imaging and clinical datasets were utilized to construct the multilayer network, with each dataset corresponding to a layer within the network. The datasets were scrutinized to address missing values and to determine which patients had data available for each layer. Notably, no imputation techniques were employed in this study. The patients were stratified into three groups: healthy, mild and severe, based on the clinical diagnosis variable.

## Multilayer network construction

Following the workflow proposed in [11], the first step in building the multilayer network was to construct individual networks from each of the six datasets by computing mutual information between nodes within each layer (mutual information was preferred over linear measures of correlation to account for the nonlinear nature of many biological processes). The networks within each layer were constructed separately, to highlight their inherent differences and to make use of the maximum number of available subjects for each dataset because not all subjects have data for all the layers. Moreover, when computing the mutual information between a pair of variables, we included only those subjects who had valid (non-missing) data for both variables in that pair. This approach allows us to maximize the use of available data for each individual mutual information estimate, rather than restricting the analysis to the subset of subjects with complete data across all modalities. Once the individual networks were constructed, features between layers were connected using mutual information too, based on the information shown in Fig 8. However, not all layers were interconnected due to a predetermined hierarchy applied to the system. This resulted in a six-layer interconnected network, with each layer comprising features derived from the original six datasets. The network construction process is illustrated in Fig 5. Furthermore, a secondary network was then created by incorporating all six datasets, employing linear correlation (Pearson coefficient) to establish the edge's directionality. This latter network was then used in the path analysis.

**Calculation of correlation for edges.** Mutual information is a nonlinear measure of the dependence between numerical variables [26]. It is not limited to variables in real numbers or to linear relationships, so the mutual information is more general than Pearson's correlation coefficient [27].

For discrete variables, mutual information is calculated using the binning method, which consists in partitioning the supports of $X$ and $Y$ into bins of finite size [28]:

$$I(X;Y) \approx \sum_{ij} p(i,j) \log\left(\frac{p(i,j)}{p_x(i)p_y(j)}\right) \tag{1}$$

If $n_x(i)$ is the number of points from $X$ that fall into the bin $i$ (analogously for $n_y(j)$), $n(i,j)$ is the number of points that fall in the intersection, and $N$ is the total number of points (sample size), then:

$$p_x(i) \approx \frac{n_x(i)}{N} \qquad p_y(j) \approx \frac{n_y(j)}{N} \qquad p(i,j) \approx \frac{n(i,j)}{N} \tag{2}$$

which leads to (1).

On the other hand, the multilayer network used for the path analysis was constructed using Pearson correlation coefficient, that provides a measure of the strength and direction of linear relationships and it is a value between -1 and 1. Both measures were computed using the *scikit-learn* library in Python [29].

**Permutation test.** We performed permutation tests to establish the statistical significance of the mutual information and the Pearson coefficient quantifiers. To that end, we randomized the data and calculated the statistics of interest for this new dataset for 1000 realizations, leading to a null distribution of the quantifier. A significance level of $\alpha = 0.05$ was chosen, meaning that the true statistics must be larger (in the case of a one-tailed test) or larger or smaller (in the case of a two-tailed test) than 95% of the randomized measures to reject the null hypothesis.

**Connectivity.** The *density* or *connectance* of a network is the fraction of observed edges to the maximum possible number of edges (without self-edges), which is $\binom{n}{2} = \frac{1}{2}|V|(|V| - 1)$ [30]. $|V|$ is the *order* or number of nodes of a network and $|E|$ is the *size* or number of edges of a network.

$$d = \frac{2|E|}{|V| \cdot (|V| - 1)} \tag{3}$$

For a weighted network we use an adaptation of the previous expression, changing the numerator to the sum of the edge weights.

$$d = \frac{2 \sum_{v \in V, u \in V, u \neq v} weight(u,v)}{|V| \cdot (|V| - 1)} \tag{4}$$

A subtype of multilayer network is the bipartite network: networks in which there are two types of nodes, belonging to two distinct subnetworks $G_i$ and $G_j$, in such a way that edges can only connect nodes of different types. For this type of network, the density has the following definition:

$$d_{bip} = \frac{|\text{edge weights between } G_i \text{ and } G_j|}{|V_i| \cdot |V_j|} \tag{5}$$

Note that in this case the maximum number of possible connections (denominator) is $|V_i| \cdot |V_j|$ because each node $i$ from $G_i$ can be connected to all nodes $j$ from $G_j$.

## Path identification

**Average shortest path length.** We used Dijkstra's algorithm to solve the single-source shortest paths problem in our weighted graphs. This algorithm computes a minimal spanning tree, a tree-like structure that connects the source node (the initial node, in which the path starts) to every other node in the graph following the shortest path to each one.

This method can be used to measure a statistic of interest for characterising a network. In particular, the average shortest path length is given by the formula (6):

$$\alpha = \frac{\sum_{v,u \in V} d(v,u)}{|V|(|V| - 1)} \tag{6}$$

where $V$ is the set of nodes, $|V|$ is the total number of nodes and $d(v,u)$ is the length of the shortest path between $v$ and $u$. In the case of networks in which the weight associated with the connections is not the distance that separates the nodes, but how strong the connection between them is, we take the distance as the inverse of this weight. Then, for a weighted network, the average shortest path length gives information on how strong the connections between nodes are.

**Boolean modeling.** As described before [11], the method of path identification involved constructing a combined six-layer network using Pearson correlation. Inspired by [31], Boolean simulations were then employed to analyze the flow of information across the network, with a particular focus on how perturbations affect nodes in different layers, especially those related to the phenotype. The aim was to identify variations in paths that converge in the clinical phenotype in individuals with AD.

Each element in the network, belonging to one of the six layers, is assumed to be either active or inactive. The Boolean simulation starts in a random state, where each element has a 50% chance of being active or inactive. At each iteration, the activation status of the elements is updated based on the sum of their neighbors' states (Fig 1). The connections between the elements are either activating (positive) or inhibitory (negative). To determine whether a node will be active or inactive in the next iteration, each neighbour contributes a score based on the weight and sign of the corresponding Pearson correlation. The total sum of the weights of the neighbours determines the node's activation status in the subsequent iteration.

The simulation was conducted for 100 steps by updating the states of the elements in each iteration. One node was selected as the input and manually switched between active and inactive states in a defined period (i.e., 10 iterations active, then 10 iterations inactive) to analyze how perturbations propagate through the network and impact a given phenotype (output). To account for the stochastic nature of biological systems and prevent the simulation from settling into a fixed state, noise was introduced by assigning a probability for each element to change its state at each iteration. Fig 2 illustrates the effect of noise on the system. A noise level of 5% was chosen as it highlights differences in the cross-correlation of signals between nodes. In the absence of any noise, many nodes remain inactive or active for most of the simulation, resulting in high cross-correlations between nodes and masking the subtle variations in connection strength.

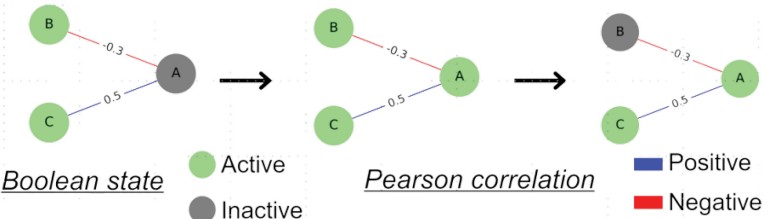

**Fig 1. Boolean dynamics**. Boolean dynamics are implemented on the networks, wherein the activation state of nodes undergoes changes determined by the cumulative sum of edge weights from their direct neighbors, taking into account the signs of connections as indicated by the Pearson coefficient: if the total sum is positive, the node becomes active, otherwise it becomes inactive. In this example, node A starts out inactive. Node B is active and is linked to node A by an edge with a negative contribution of 0.3. On the other hand, node C, which is also active, contributes to the activation of A positively with a weight of 0.5. In total, the contribution is 0.2, so node A is activated. In the next step, A is active and contributes positively to the activation of C, so C remains active. However, the contribution to node B is negative, so it becomes inactive.

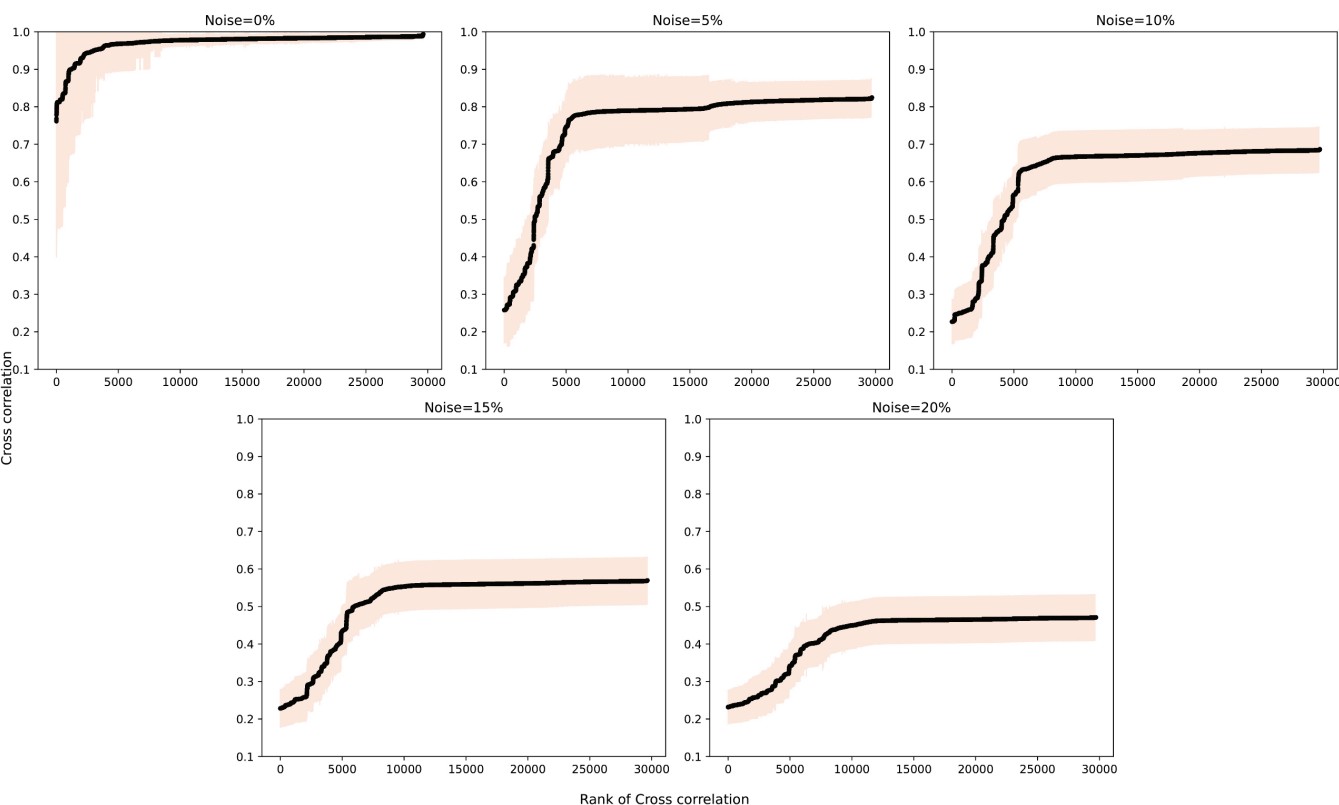

**Fig 2. Impact of noise on the cross-correlation coefficient of signals between nodes in a combined network**. In the absence of noise (0%), the majority of cross-correlation values approached 1, making it challenging to rank node pairs based on connection strength. However, with 5% noise, the cross-correlation values exhibited greater variation, enabling easier identification of paths between a selected source and target.

Following the simulations, a temporal cross-correlation function was computed between all pairs of nodes using the same measure of similarity as in [31]. The highest cross-correlation, potentially occurring at a non-zero lag time, was identified, and its reciprocal was assigned as a weight to the edges within the existing network. The high correlation values corresponded to low weights, and if there was no edge in the original network, no edge was added to the new network. The target phenotype was chosen, and the most efficient paths between it and the fixed source were identified based on the lowest path score, which was defined as the total sum of the weights (inverse maximum cross-correlations) of the edges connecting the source and target (Fig 3). The path score prioritized paths with both a minimal number of steps and high cross-correlations between nodes within the path. Dijkstra's algorithm was employed to pinpoint paths with the lowest path scores. Simulations were carried out for every conceivable pair of inputs and outputs to investigate the flow of information throughout the entire network, offering insights into the underlying pathology in AD.

To test the consistency of the results, we performed negative controls of the paths by permuting the six-layer network constructed with the Pearson correlation, as illustrated in Fig 4. This process involved swapping edges between node pairs in the network, and it was repeated 100 times. Importantly, an edge swap was only performed if it did not result in two edges connecting the same node pair. This approach maintained the original network's degree distribution. Additionally, the weights associated with each edge were permuted. The code to perform the permutation of the edges has been modified from the code published on [32]. Each edge

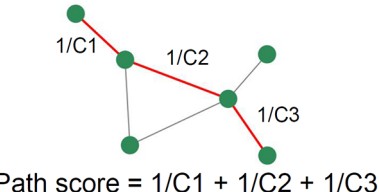

Path score = 1/C1 + 1/C2 + 1/C3

**Fig 3. Calculation of the path score**. The cross-correlation coefficient is computed by assessing the signals for each connected node pair. A path score is then determined for all possible paths, defined as the sum of the reciprocals of the cross-correlation coefficients between consecutive node pairs along a specific path.

in the original network underwent this edge swapping technique 10 times. Once the permutations were completed, the top paths for each network were identified using the same method as before (Fig 5).

## Results

A cross-sectional study was performed by integrating data from the ADNI cohort (https://adni.loni.usc.edu/) at different biological scales: MRI, PET scans (including A$\beta$, tau and FDG PET), CSF/plasma proteomics, genetic information, risk factors and cognitive tests and diagnosis assessment conforming the phenotype (Fig 6). The results center around the identification of paths connecting these biological layers. The following paragraphs outline how these paths were discovered, and which sources are more strongly linked to the phenotype. The initial step provides descriptive details about the data, followed by the construction of the networks. Subsequently, Boolean simulations are executed, and ultimately, the most significant paths and nodes are chosen.

### Comprehensive phenotypic profiling: multi-omics, imaging, and clinical data across the ADNI cohort

The subjects for the study were classified, using exclusively clinical criteria, as controls (n=622), subjects with MCI (n=807), or subjects with AD (n=533). As shown in Table 1, the mean age of the 3 groups was equivalent at approximately 74 years. There were an approximately equal number of men and women in the control, but there were more men than women in the MCI and AD groups. More than half of the patients in the AD group were *APOE4* carriers, while the controls were less than 30%. Data corresponding to the concentration of cerebrospinal fluid biomarkers A$\beta_{42}$, tau and p-tau were collected from the University of Pennsylvania Alzheimer's Disease Clinical Core dataset.

### Multilayer networks in AD

In order to create networks for each of the six layers, connections between pairs of elements within each layer were established using mutual information, as explained in [11]. Node pairs within the same layer were connected to one another with a weight equal to the normalized mutual information between them. A statistical threshold was implemented to examine whether the correlation for a given pair was strong enough to establish an edge. Specifically, a node pair's actual mutual information value was compared to a surrogate distribution of mutual information values derived from random permutations of the data, and a p-value was computed. Only pairs with $p<0.05$ were retained as significant, and an edge was established between the corresponding nodes.

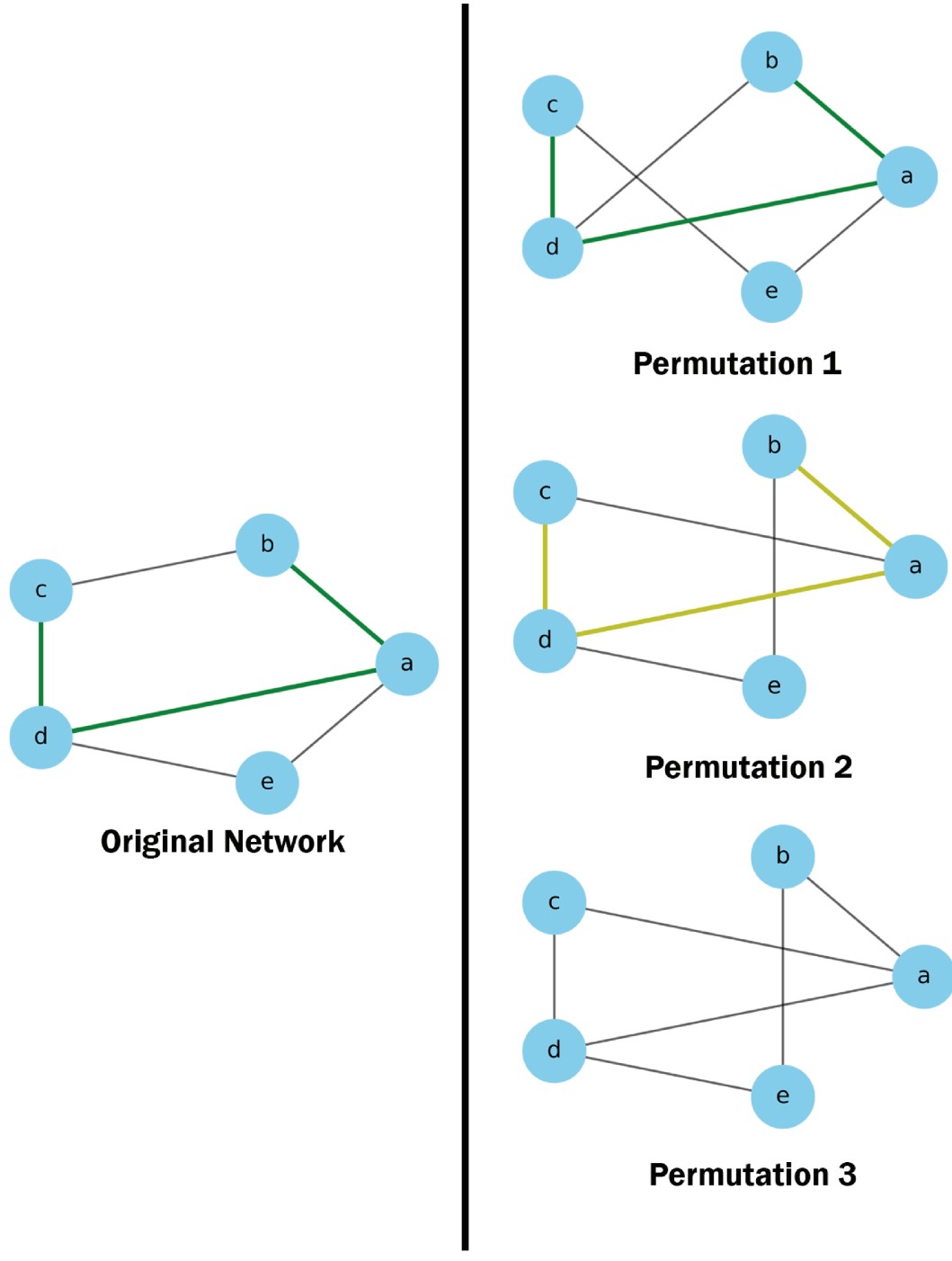

**Fig 4. Network permutation for negative controls**. In this example, in the first permutation, the edges (b,c) and (d,e) were exchanged with the edges (b,d) and (c,e). For the second permutation, the edges (a,e), (b,c) were swapped with the edges (a,c) and (b,e). In the third permutation, the top network's edge swap was applied first, followed by the middle network's edge swap: (a,b), (b,c) and (d,e) were exchanged for (a,c), (b,d) and (b,e). Three possibilities were considered when determining if the paths from the original network appeared in the permuted networks. In the first permutation, the path existed in the permuted network and was also identified as a top path. In the second permutation, the original path existed in the permuted network but was not identified as a top path. Finally, in the third permutation, the original path did not exist in the permuted network at all.

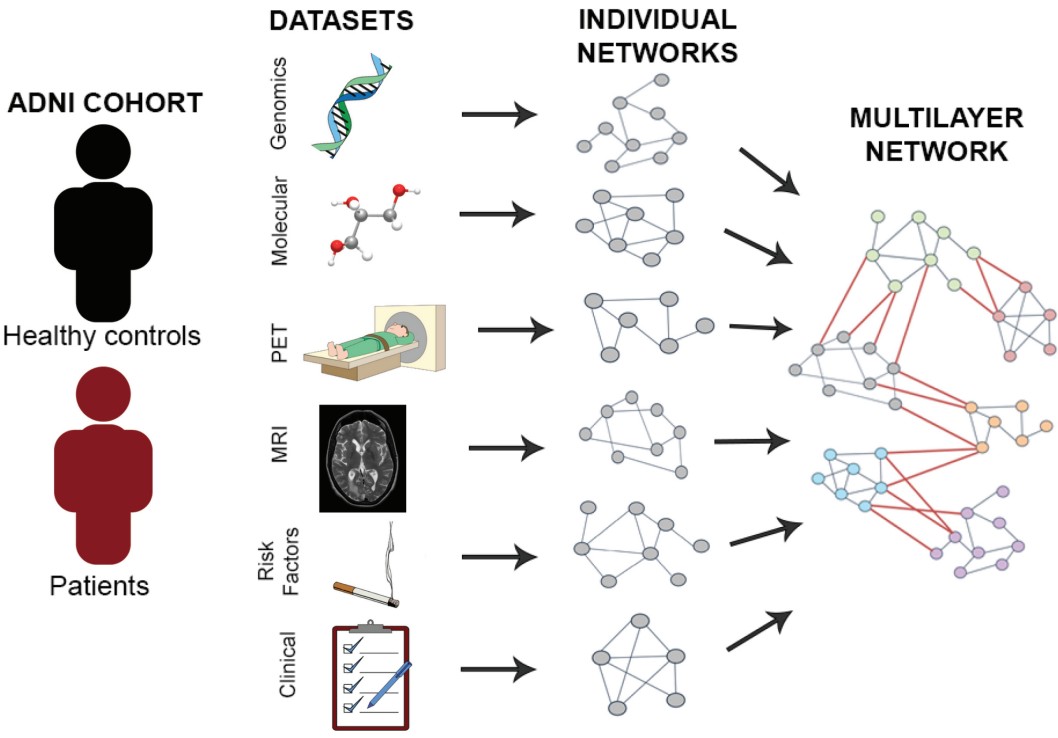

**Fig 5. Illustration of the multilayer network construction**. Icons were adapted from freely available resources at Openclipart and Wikimedia Commons, under Creative Commons licenses.

The individual genetic, molecular and phenotype networks have few variables, due to prior selection performed to retain only those features thought to have an appreciable effect on the diagnosis of AD (Fig 6). Genetic and phenotypic networks are less connected than the molecular network: the connections in the latter have a higher associated weight. In turn, the PET and MRI networks are larger in size due to the numerous brain regions contained in the dataset. The nodes in these networks represent functionally connected regions, reflecting relationships in terms of metabolic processes or neuronal activity. On the other hand, the risk factor network also has a large number of nodes, but the weight of their connections is generally low, which gives us an idea of how little dependence there is between the variables. This is to be expected given the nature of this dataset: risk factors range from the patient's gender to whether they have respiratory problems, therefore such weak correlations between variables are not surprising.

Our approach to the multilayer network hierarchy is originally hypothesis-based, rather than data-based: each individual network forms a layer or level in the structure of this model, representing a different biological scale (Fig 7). The genomics layer sits at the bottom of the hierarchy and is connected with the molecular layer, constituting the microscopic substrate of the disease. The tissue imaging layers (PET and MRI) are the central functional layers, which, connected to each other and forming a bridge between the microscopic (genome and molecules) layer and the clinical output. The risk factor nodes form a loosely connected cloud around the main structure, although some of their connections are strong: this is the case of the year of birth (age) and gender of the patients. This analysis thus produced a network of six connected layers, where each layer contains features (variables) from each of the six original

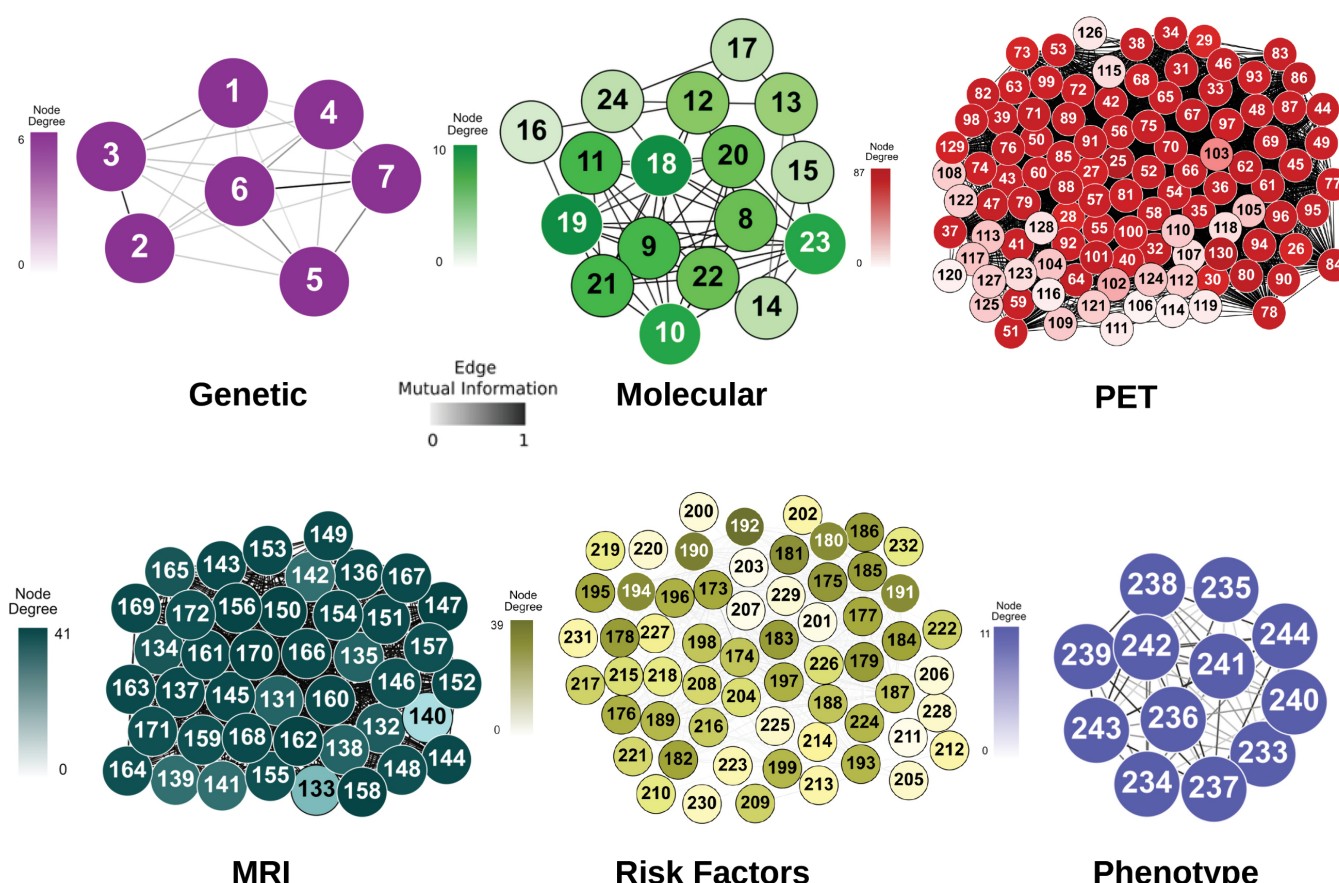

**Fig 6. Individual networks**. The data from each layer is taken from the ADNI cohort and used to create networks, with nodes representing the dataset's elements (genetics, molecular, PET, MRI, risk factors, and phenotype) and edges representing the mutual information between element pairs across all subjects. For clarity, nodes are labeled numerically in the figure, and the corresponding variable names are provided in S1–S6 Tables. See high resolution networks at https://dsb-lab.github.io/networks/

**Table 1**. Characteristics of the ADNI cohort

| Groups | Controls (n=622) | MCI (n=807) | AD (n=533) |
|---|---|---|---|
| Age | 74 ± 7 | 74 ± 8 | 75 ± 8 |
| Female, % | 54.5 | 39.1 | 43.6 |
| Age at disease onset | – | – | 76 ± 8 |
| APOE $\epsilon4$ carriers, % | 29.4 | 48.0 | 64.8 |
| A$\beta_{42}$, pg/ml | 203 ± 50 | 174 ± 53 | 143 ± 41 |
| tau, pg/ml | 65 ± 31 | 86 ± 52 | 122 ± 60 |
| p-tau, pg/ml | 32 ± 18 | 38 ± 21 | 50 ± 29 |

datasets (Fig 7). Here, connections between layers that do not conform to the proposed structure have been eliminated, as the hierarchy that we apply to the system is for representation only. In the rest of the paper we consider the global network without a predefined hierarchy, to avoid possible biases.

First we proceeded to examine whether the connectivity of the global network reflects the biological organization into layers discussed above. To that end, node pairs were connected to each other irrespective of their layer, without a predefined hierarchy, and the *connectance*

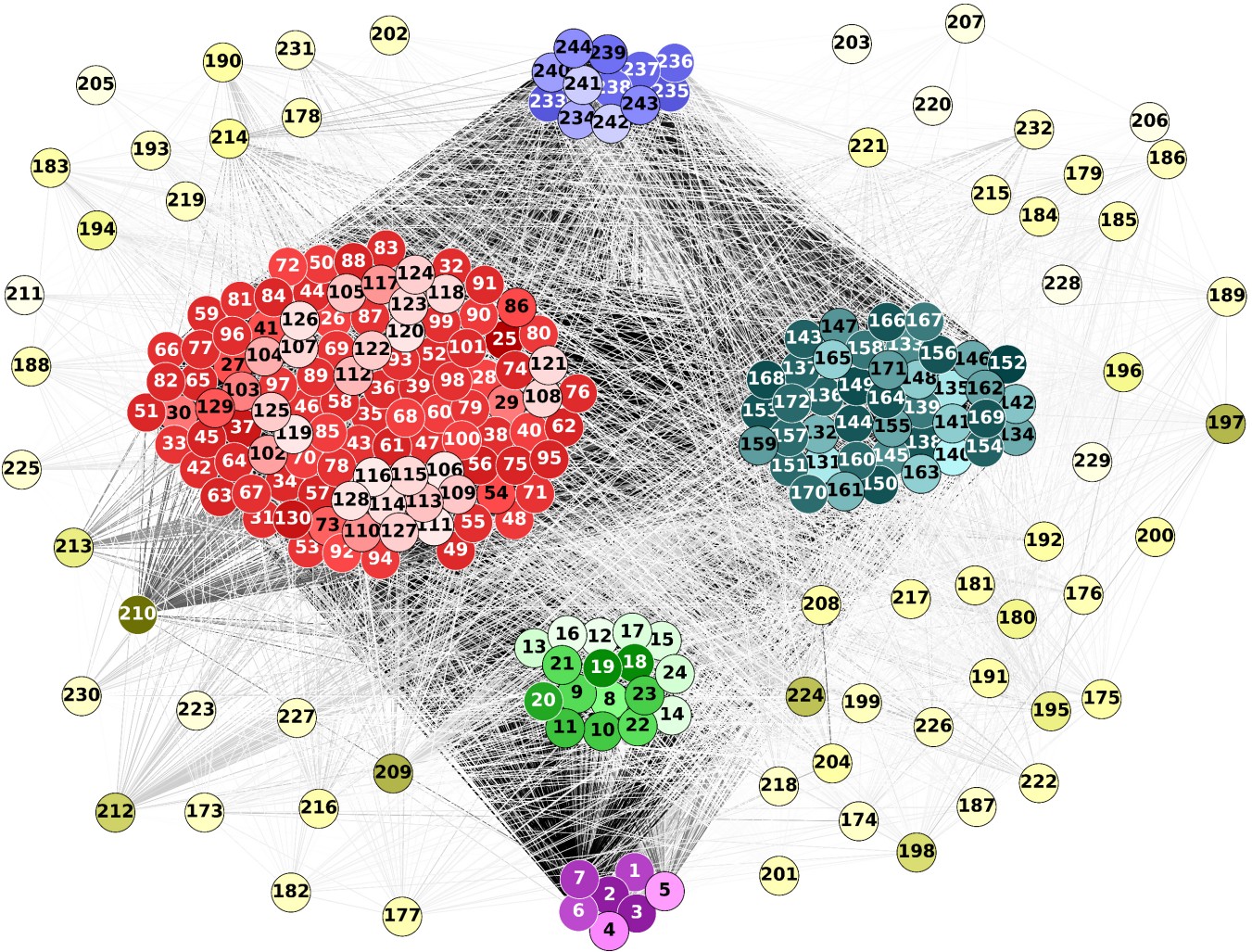

**Fig 7. Multilayer network**. Following a hierarchy that connects each layer successively, starting with the genomics layer and working up to the phenotypic (clinical) layer, individual networks are created and linked together using mutual information once again. The risk factors nodes are shown in the periphery of the network, as well as being external to the biological hierarchy, for visualization purposes. For clarity, nodes are labeled numerically in the figure, and the corresponding variable names are provided in S1–S6 Tables. See high resolution image at https://dsb-lab.github.io/multilayer_net/.

*matrix* between two layers *i* and *j* was calculated as follows:

$$C_{ij} = \begin{cases} d_i, & \text{if } i = j \\ d_{ij}^{bip}, & \text{if } i \neq j \end{cases} \tag{7}$$

Here *d* is the standard connection density within a layer, as defined by (4) in the Materials and Methods section, and $d_{bip}$ is the connection density of the bipartite network where each of two layers is one of the two node sets, as defined by (5). The latter measure is used because in the case of off-diagonal elements (different layers), we are only interested in how connected the nodes of one layer are to those of the other, but not with nodes of their own layer.

The resulting connectance matrix is shown in Fig 8. A significant level of network modularity was discovered by this comparison of connections within and between layers,

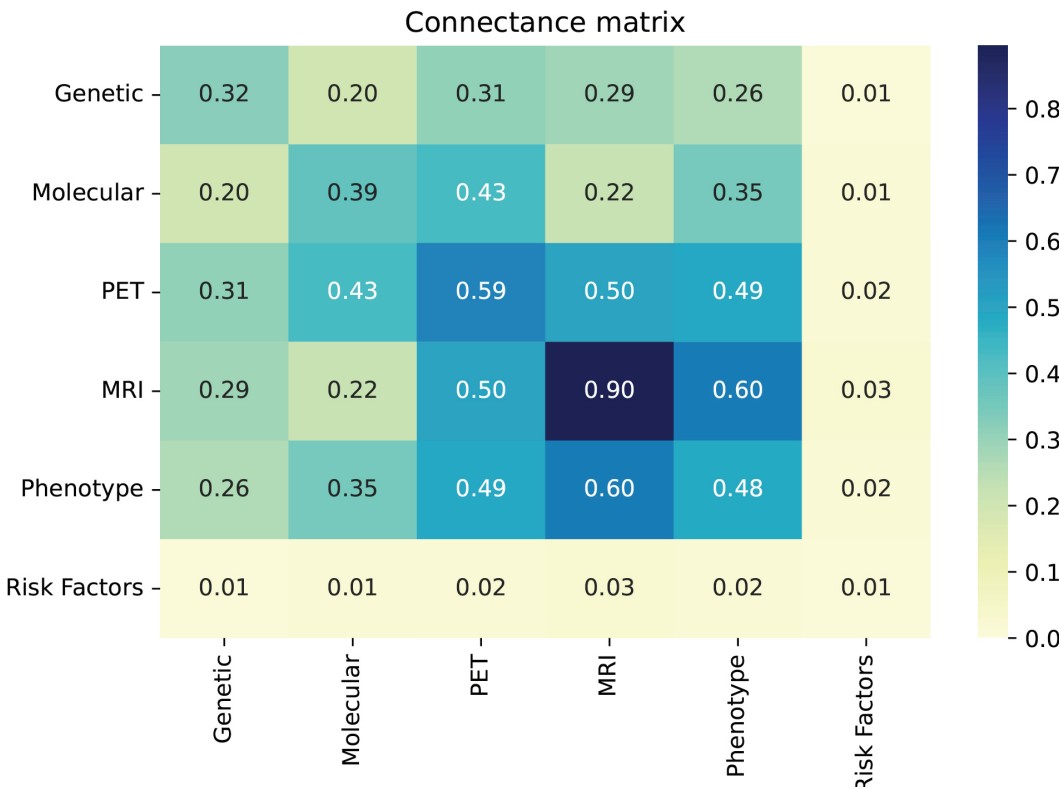

**Fig 8. Network densities within and between layers**. The connectance matrix was calculated using the expression in (7).

confirming the existence of an underlying multilayer structure: the connectivity within individual layers is higher than that between different layers. With the exception of risk factors, features within a level tend to be more correlated than between levels. The most prominent example of this is the MRI layer. The nodes in this layer are so closely related to each other because of the similarity between all the variables, representing levels of atrophy in different parts of the brain, as opposed to, for example, the nodes in the molecular layer which mostly represent concentrations of different proteins.

## Dynamic network analysis identifies paths associated with phenotype

To obtain a functional view of the information flow across layers, we aimed to integrate all six layers into paths that reflect network dynamic interactions. We constructed a single network comprising all layers using linear (Pearson) correlations that distinguish between stimulatory or inhibitory edges, depending on the correlation value being positive or negative, respectively. To explore the logical structure underlying the network, we conducted logic (Boolean) simulations. These simulations use knowledge of activating and inhibiting relationships between nodes while ignoring the exact functional reactions between the nodes, thus providing a qualitative description of the system [31]. Nodes are either active or inactive, and their states are updated synchronously in each iteration of the simulation, depending on the activation states of their direct neighbors and the weights of the corresponding connections.

Our next objective was to investigate how dynamic changes in a specific input propagate through the network and ultimately impact a given phenotype. We achieved this by

performing Boolean simulations, as described above, where the input node was periodically switched between active and inactive states. The responses of all nodes in the network were then measured by computing the temporal cross-correlation function between their time-varying state and the dynamic input signal. We then identified the paths in the network with the highest overall temporal cross-correlation between their signals, which indicate how information flows from the input to the output. The paths were chosen based on the lowest path score, which was defined as the total sum of the weights (inverse maximum cross-correlations) of the edges connecting the source and target [31]. While these paths may not necessarily reflect physical interactions among nodes, they represent groups of nodes that co-vary statistically more strongly with each other than the rest of the network.

For each combination of inputs and outputs, we selected the top ten paths with the highest joint cross-correlation values between their constituent nodes, resulting in a total of 30,000 combinations. Figs 9, 10, 11, 12 and 13 show these paths for the five inputs (genetic,

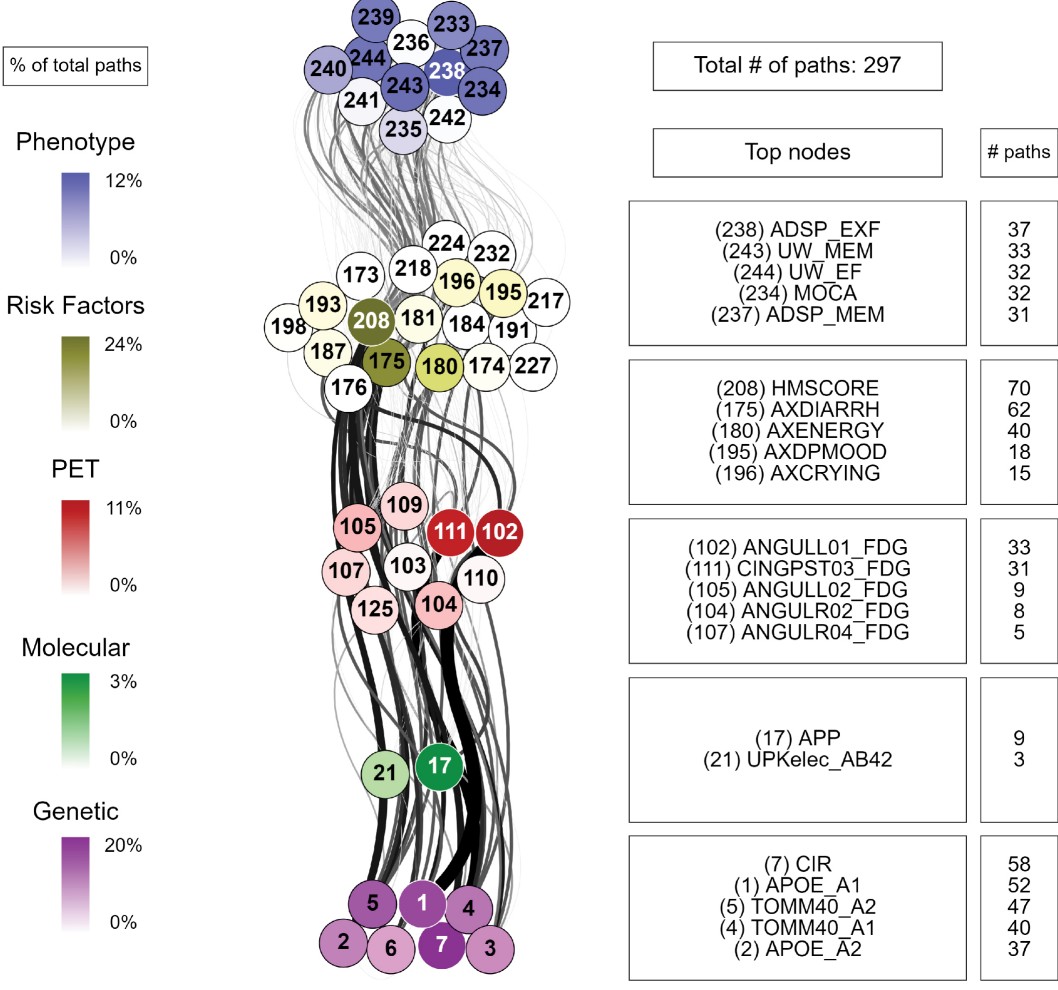

**Fig 9. Path analysis from the genetic layer in ADNI participants**. Depictions of the multilayer paths identified through Boolean simulations with the genetic layer as the starting point. The top paths, meeting criteria for negative controls, are presented for each input (genetic) - output (clinical phenotype) pair. Nodes within each layer are color-coded to reflect the node's degree, indicating the frequency of its appearance in a path as a percentage of the total paths. For clarity, nodes are labeled numerically in the figure, and the corresponding variable names are provided in S1–S6 Tables. For detailed high-resolution paths, please refer to https://dsb-lab.github.io/network_paths/.

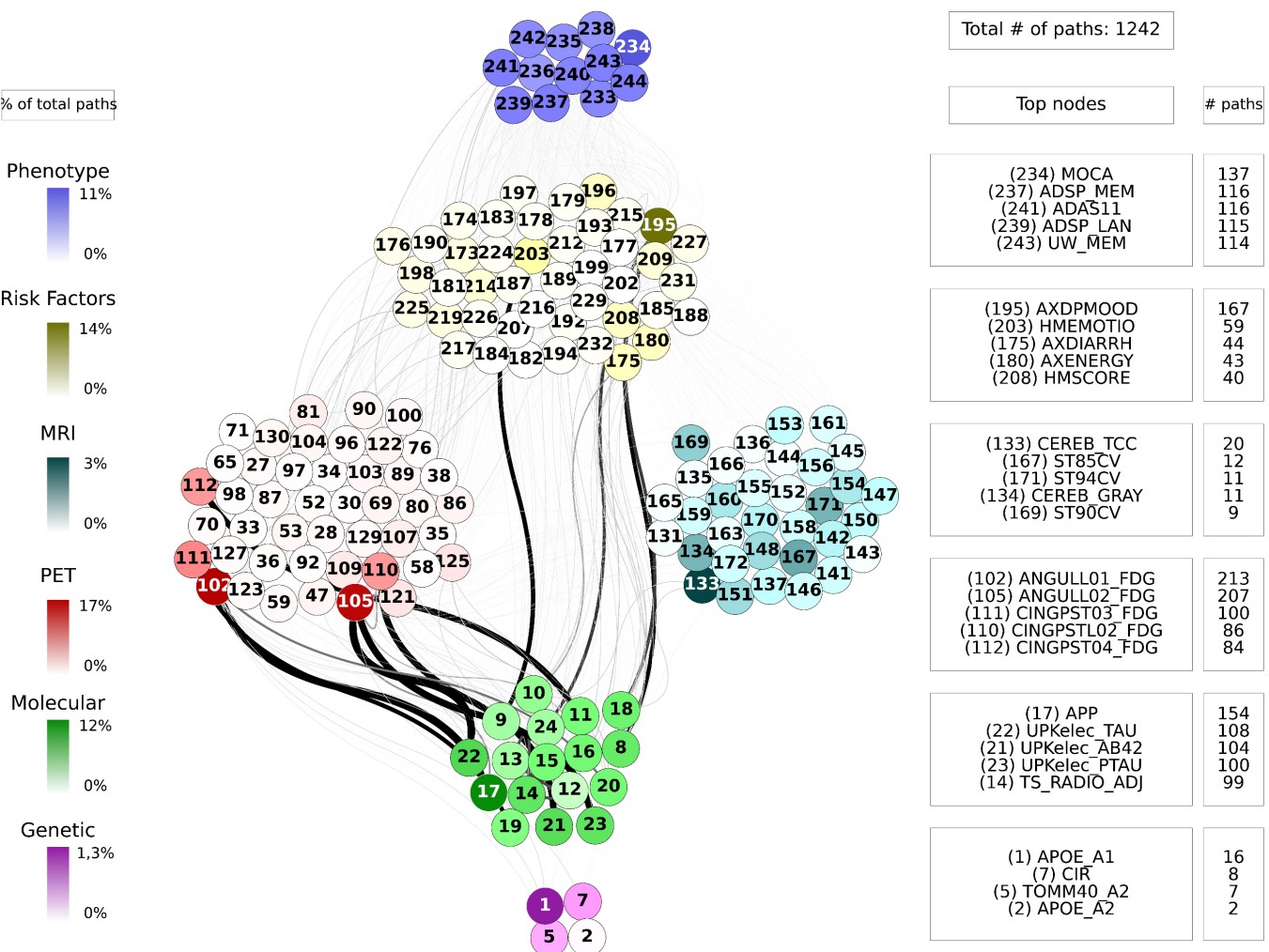

**Fig 10. Path analysis from the molecular layer in ADNI participants**. Depictions of the multilayer paths identified through Boolean simulations with the molecular layer as the starting point. The top paths, meeting criteria for negative controls, are presented for each input (molecular) - output (clinical phenotype) pair. Nodes within each layer are color-coded to reflect the node's degree, indicating the frequency of its appearance in a path as a percentage of the total paths. For clarity, nodes are labeled numerically in the figure, and the corresponding variable names are provided in S1–S6 Tables. For detailed high-resolution paths, please refer to https://dsb-lab.github.io/network_paths/.

molecular, PET, MRI and risk factors) and outputs (phenotype) pairs for participants in the cohort. Darker color represents more connections among the nodes.

To evaluate the specificity of the Boolean simulations, we randomly permuted the network connections as described in the Materials and Methods section, to identify negative control paths that were then compared to those identified in the original networks. We focused on paths that appeared in less than 1% of the permutations. Out of the 30,000 total paths identified from participants in the cohort, 17,877 did not appear at all in 100 realizations of the simulations in the permuted paths. The top paths (those that passed the test for negative controls) are shown in Figs 9, 10, 11, 12 and 13 for each input (genetic, molecular, PET, MRI and risk factors) and output (phenotype) pair. All variables with their acronyms are listed in S1–S6 Tables. Also, to better illustrate the interrelationships between phenotypic variables, we provide in S1 Fig a matrix for each path analysis with the number of connections between

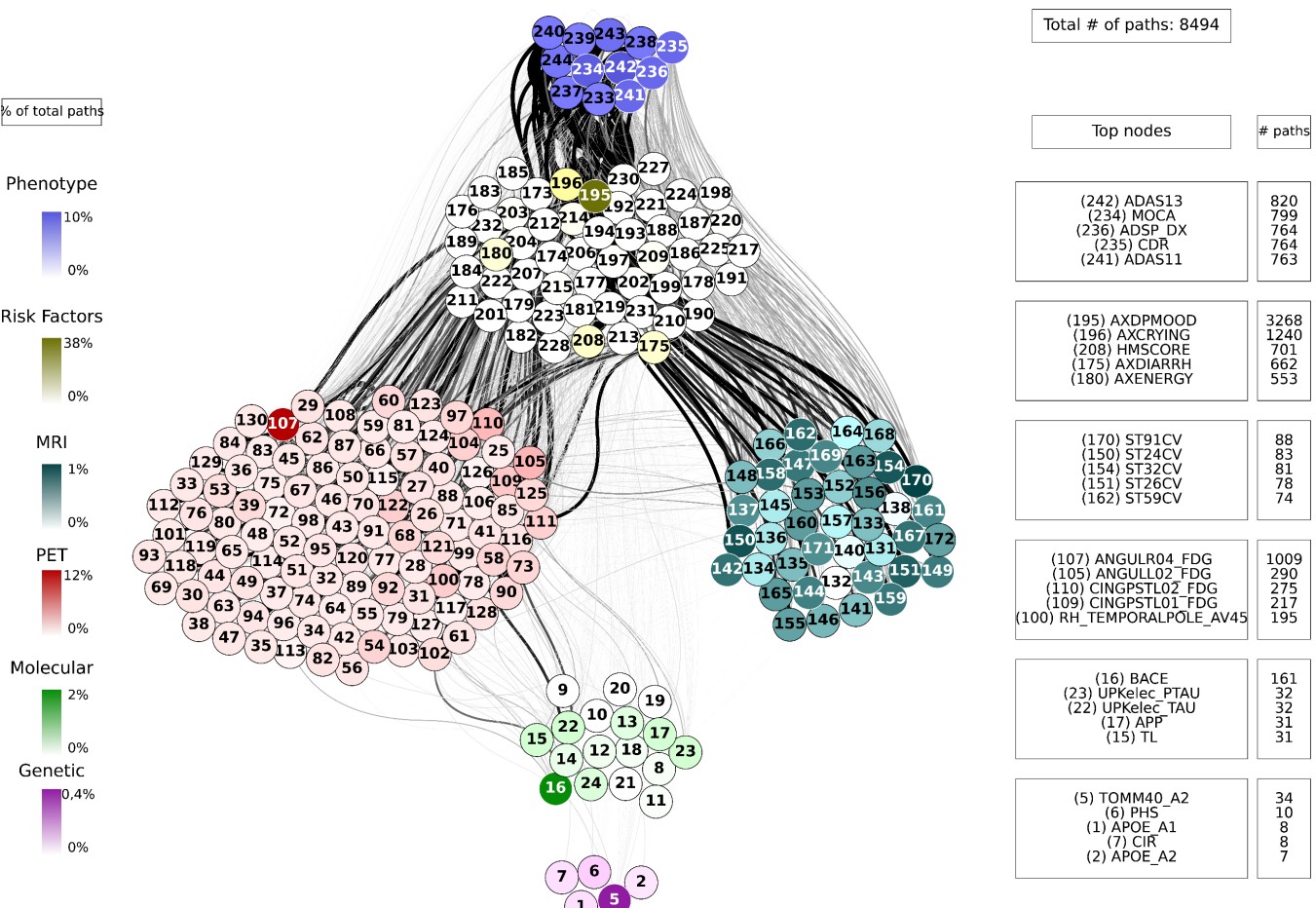

**Fig 11. Path analysis from the PET layer in ADNI participants**. Depictions of the multilayer paths identified through Boolean simulations with the PET layer as the starting point. The top paths, meeting criteria for negative controls, are presented for each input (PET) - output (clinical phenotype) pair. Nodes within each layer are color-coded to reflect the node's degree, indicating the frequency of its appearance in a path as a percentage of the total paths. For clarity, nodes are labeled numerically in the figure, and the corresponding variable names are provided in S1–S6 Tables. For detailed high-resolution paths, please refer to https://dsb-lab.github.io/network_paths/.

the nodes of the phenotypic layer. The method for path identification and network permutation is illustrated in the Materials and Methods section (see Fig 4). To further explore heterogeneity and disease progression, we performed separate path analyses within the control, mild cognitive impairment (MCI), and Alzheimer's disease (AD) groups using the same methodology described above. For each diagnostic subgroup and input data layer, we identified the top 10 paths with the highest joint cross-correlation. S8–S22 Tables provide these top paths along with the frequency of co-occurring node pairs within each path. This stratified analysis reveals both common and unique network patterns across disease stages, offering insight into the evolving phenotypic complexity of AD.

## Path analysis

We now discuss the top 10 paths with the highest joint cross-correlation values. In what follows, the information flows along each path from left to right, and the perturbation starts at the first node. The paths more commonly found are the following.

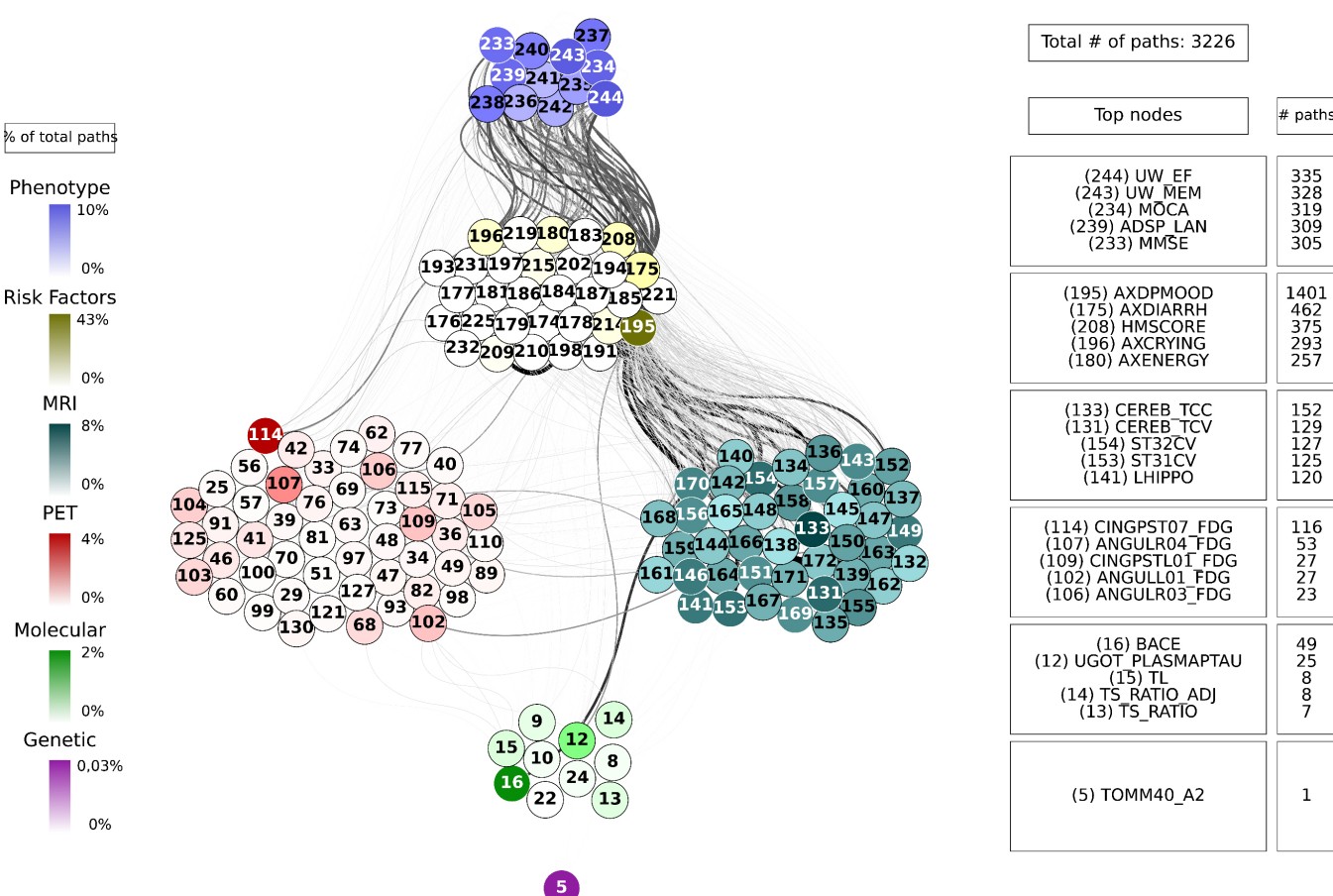

**Fig 12. Path analysis from the MRI layer in ADNI participants**. Depictions of the multilayer paths identified through Boolean simulations with the MRI layer as the starting point. The top paths, meeting criteria for negative controls, are presented for each input (MRI) - output (clinical phenotype) pair. Nodes within each layer are color-coded to reflect the node's degree, indicating the frequency of its appearance in a path as a percentage of the total paths. For clarity, nodes are labeled numerically in the figure, and the corresponding variable names are provided in S1–S6 Tables. For detailed high-resolution paths, please refer to https://dsb-lab.github.io/network_paths/.

- APOE_A1 (copy 1 of *APOE* gene) → ANGULL01_FDG (globally normalized CMRgl from left angular gyrus) → AXRASH (rash) → any node of the phenotype layer, when the input is applied to the genetic layer (Fig 9).
- UPKelec_TAU (CSF total Tau using the fully automated Roche Elecsys immunoassay) → ANGULL02_FDG → APP (CSF amyloid precursor protein) → ADSP_VSP (harmonized composite visuospatial score), ADSP_EXF (harmonized composite executive function score), ADSP_MEM (harmonized composite memory score), MMSE (Mini Mental State Examination) and MOCA (Montreal Cognitive Assessment test), when the input is applied to the molecular layer (Fig 10).
- CINGPSTR12_FDG (globally normalized CMRgl from right posterior cingulum cortex) → AXCRYING (crying) - AXDPMOOD (depressive mood) → ADSP_EXF, when the input is applied to the PET layer (Fig 11).
- ST44CV (cortical volume of left parahippocampal) → AXDPMOOD - GDS (Geriatric Depression Score) → MOCA, when the input is applied to the MRI layer (Fig 12).

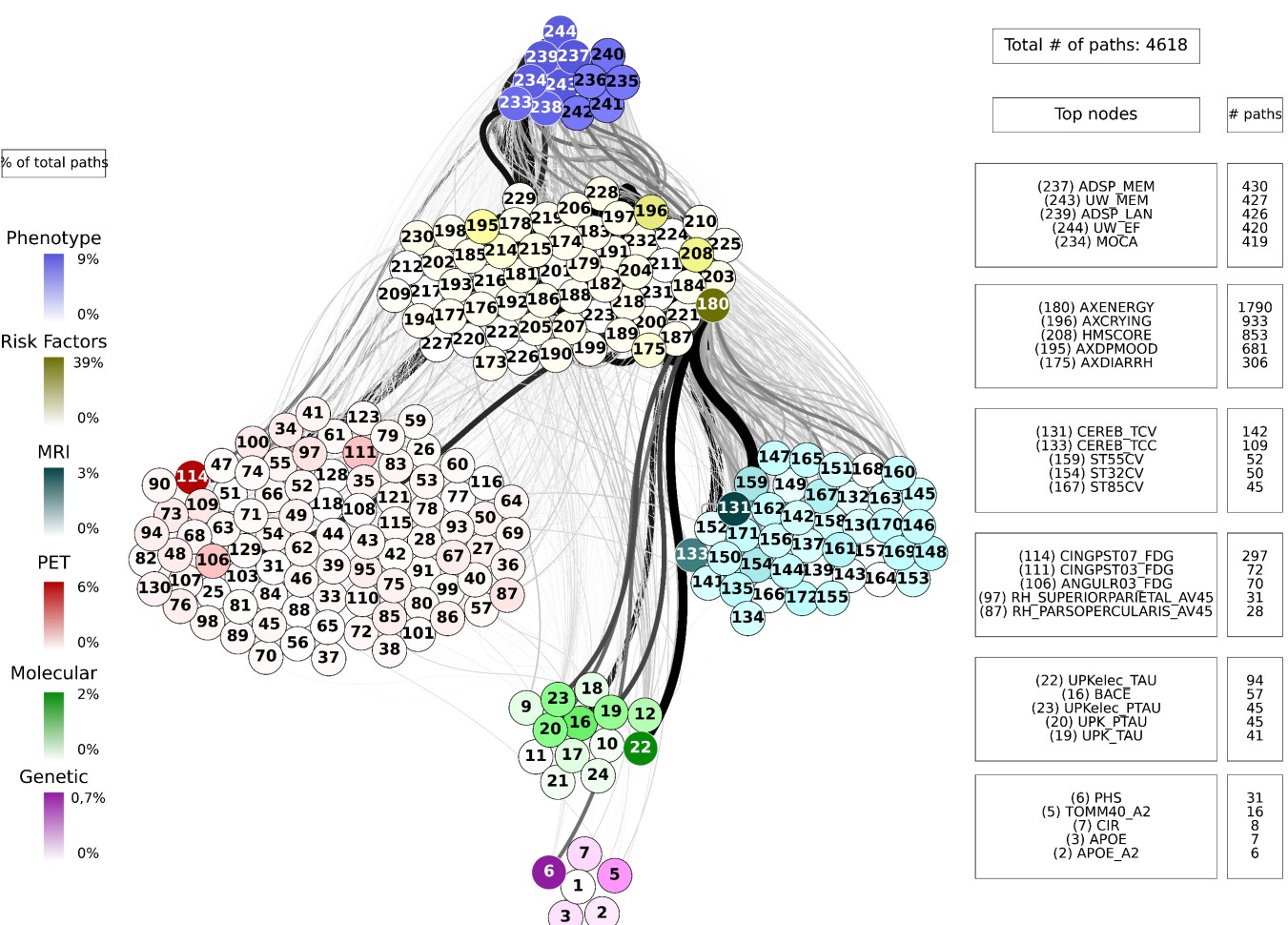

**Fig 13. Path analysis from the risk factors layer in ADNI participants**. Depictions of the multilayer paths identified through Boolean simulations with the risk factors layer as the starting point. The top paths, meeting criteria for negative controls, are presented for each input (risk factors) - output (clinical phenotype) pair. Nodes within each layer are color-coded to reflect the node's degree, indicating the frequency of its appearance in a path as a percentage of the total paths. For clarity, nodes are labeled numerically in the figure, and the corresponding variable names are provided in S1–S6 Tables. For detailed high-resolution paths, please refer to https://dsb-lab.github.io/network_paths/.

- AXELMOOD (elevated mood) - AXCRYING - AXDPMOOD → ADSP_MEM, when the input is applied to the risk factor layer (Fig 13).

When perturbing sources belonging to the genetic layer (reflecting genetic variability contributing to the risk of developing AD), we see that there is no transition through all the intermediate layers of the network: molecular nodes scarcely appear in the paths, and no node from the MRI layer appears (Fig 9). Perturbations were directly linked to the PET and risk factor layer, whose nodes represent aspects of the patient's medical history and age, and these to changes in clinical outcomes. Perturbations of the molecular network (representing changes in protein concentration) have a direct influence on the PET layer, in particular on FDG signal for the angular and cingulum posterior nodes, and are less related to impact MRI nodes, although some of these represent neurodegeneration like FDG PET (Fig 10). Then, the central involvement of FDG PET nodes suggest an important role of brain hypometabolism

in the clinical phenotype. For both genetics and molecular nodes, we found little information flow coming from the phenotype layer.

The number of paths increases considerably when we have perturbations at the imaging level (representing changes in protein concentration in the case of PET and changes in brain tissue degeneration in the case of MRI) (Figs 11 and 12) and at the risk factor level (Fig 13). Moreover, these paths show a stronger influence on the phenotype compared to those originating from, for example, the genetic layer, as visually reflected by the thicker edges in the figures. When the origin of the input is at the PET layer, the input nodes have little connectivity with the genetic layer nodes. In fact the PET nodes are mainly related to the risk factor layer and, through it, to the MRI nodes or directly to the phenotype nodes. The case of MRI is analogous: there are not so many connections with the deeper layers (genetic and molecular). The importance of the MRI nodes is more distributed, i.e. they appear with similar frequency in the paths, while we see that the relationship with the PET layer is quite centred on the nodes: CMRgl of the angular gyrus and posterior cingulum cortex. The indirect interaction with risk factors is also notable, highlighting some specific nodes related to symptomatology, which are discussed below. This indirect interaction is also found in the opposite direction, when the source belongs to the risk factor layer. In this case, however, we do not see a strong influence on the genetic layer. This is consistence with expectations: genes can be expected to influence the occurrence of some risk factors, but logically these factors cannot change the genetic information of the person, which is determined from birth.

In general, our results show that as we move towards the highest-scale layers (the imaging layers and risk factors layer), the paths have very little tendency to return to the genetic and molecular layers, which shows the importance of considering AD as a multiscale disease, with the layers connected with different strengths and where information flows from the genetic and molecular layers, having increasing influence on the phenotype as we move up the layers.

Contrary to what might be expected, the concentrations of A$\beta$ and tau do not appear frequently in the paths, thus their influence on the system is very low. It is also important to highlight the presence of the angular gyrus and posterior cingulum cortex FDG nodes from the PET layer in most of the paths, almost independently of the layer to which the source belongs. All together, our analysis points to a relevant role of brain hypometabolism (FDG PET) on the information flow compared with A$\beta$ and tau nodes.

To further characterize the role of FDG PET nodes within the network, we systematically analyzed their upstream and downstream connections with variables from other layers and showed the summary in Table 2. This analysis revealed that FDG PET nodes maintain a high number of edges with the MRI layer, with an average mutual information of $\sim 0.99$, suggesting a very strong correspondence between regional metabolic activity and structural neurodegeneration and also connect extensively with the molecular layer (average MI $\sim 0.97$), reflecting their close relationship with protein concentrations. Connections with the phenotypic layer were also robust (average MI $\sim 0.78$), highlighting the downstream influence of FDG PET features on cognitive and clinical outcomes. By contrast, FDG PET nodes showed

**Table 2**. Summary of FDG PET connectivity across network layers.

| Layer | Average Mutual Information (MI) | Number edges |
| --- | --- | --- |
| Molecular | 0.9675 | 60 |
| MRI | 0.9911 | 286 |
| Risk Factors | 0.1819 | 148 |
| Phenotype | 0.7788 | 129 |

a large number of connections with the risk factor layer, but the average MI was much lower ($\sim 0.18$), indicating weaker but widespread associations driven by the diversity of variables in this layer. Overall, this analysis, complementing the path analysis, demonstrates that FDG PET nodes are highly connected across all layers of the network, acting as central integrators of upstream molecular and structural alterations and downstream clinical phenotypes. A detailed list of FDG PET connections with nodes from all other layers is provided in S7 Table.

Among the variables in the phenotypic layer, MOCA consistently emerged as a recurrent node across all inferred paths, regardless of the input source (genetic, molecular, imaging, or risk factors). This suggests that MOCA is one of the most frequently involved phenotypic outcomes in the network, acting as a common downstream element influenced by multiple biological and clinical factors. Notably, this centrality can be attributed to MOCA's high level of correlation with a broad range of variables from other layers. The path selection process is based on dynamic Boolean simulations and quantified using temporal cross-correlation between the input signal and the state of each node. Because MOCA exhibits strong temporal cross-correlations with many other variables across the network, it is more likely to appear in high-scoring paths. Also noteworthy is the influence of the patient symptoms, whose variables are found in the risk factor layer and include low energy, diarrhea, crying, elevated and depressed mood, among others. Several of these symptoms (in particular, HMSCORE, AXDIARRH, AXENERGY, AXDPMOOD, and AXCRYING) appear recurrently across the inferred paths, regardless of the data layer from which the path originates. This is the same case as the MOCA variable.

The analysis shown above could provide information on the modifiable risk factors that can be used in preventive lifestyle modification trials. In order to detect at which levels these factors have an impact, we have depicted in Fig 14 the 20 shortest paths that arise when the origin is a risk factor node. The main sources for these paths are drowsiness, hypertension, crying, cardiovascular history and musculoskeletal pain. It can be seen that only the risk factor layer is involved and is directly related to the target, which is the phenotype layer. In particular, other risk factor nodes related to the sources appear, for example, low energy (180), related to drowsiness and muscle pain, or depressive mood (195), related to crying. No nodes from the genetic, molecular or imaging layers appear in these paths. Vascular health and muscle pain are related to executive and visuospatial function, while variable crying is also related to executive function and memory.

Consistent with the results obtained from the analysis including all participants, we observed that across all three groups (controls, MCI and AD patients) the largest number of inferred paths originates from perturbations applied to nodes in the PET and risk factor layers, compared to inputs from the genetic or molecular layers. A striking finding was that in both the MCI and AD groups, every single path, regardless of its source, includes at least one node from the PET layer, underscoring the dominant role of PET abnormalities in mediating network-level influences in more advanced stages of the disease. This pattern was not observed in the control group, where PET involvement was less frequent. The critical importance of PET features in MCI and AD likely reflects their ability to capture early metabolic and neurodegenerative changes that integrate upstream molecular signatures and are closely linked to downstream phenotypic manifestations.

Another key difference between the diagnostic groups was the presence of molecular biomarkers of Alzheimer's pathology in the paths. While in the global analysis (using all participants) the classical protein biomarkers $A\beta$ and tau appeared infrequently, this pattern changed notably in the MCI and AD groups. In these groups, concentrations of $A\beta42$ were present in a substantial proportion of the inferred paths, indicating that their influence becomes more pronounced in individuals already presenting cognitive impairment or

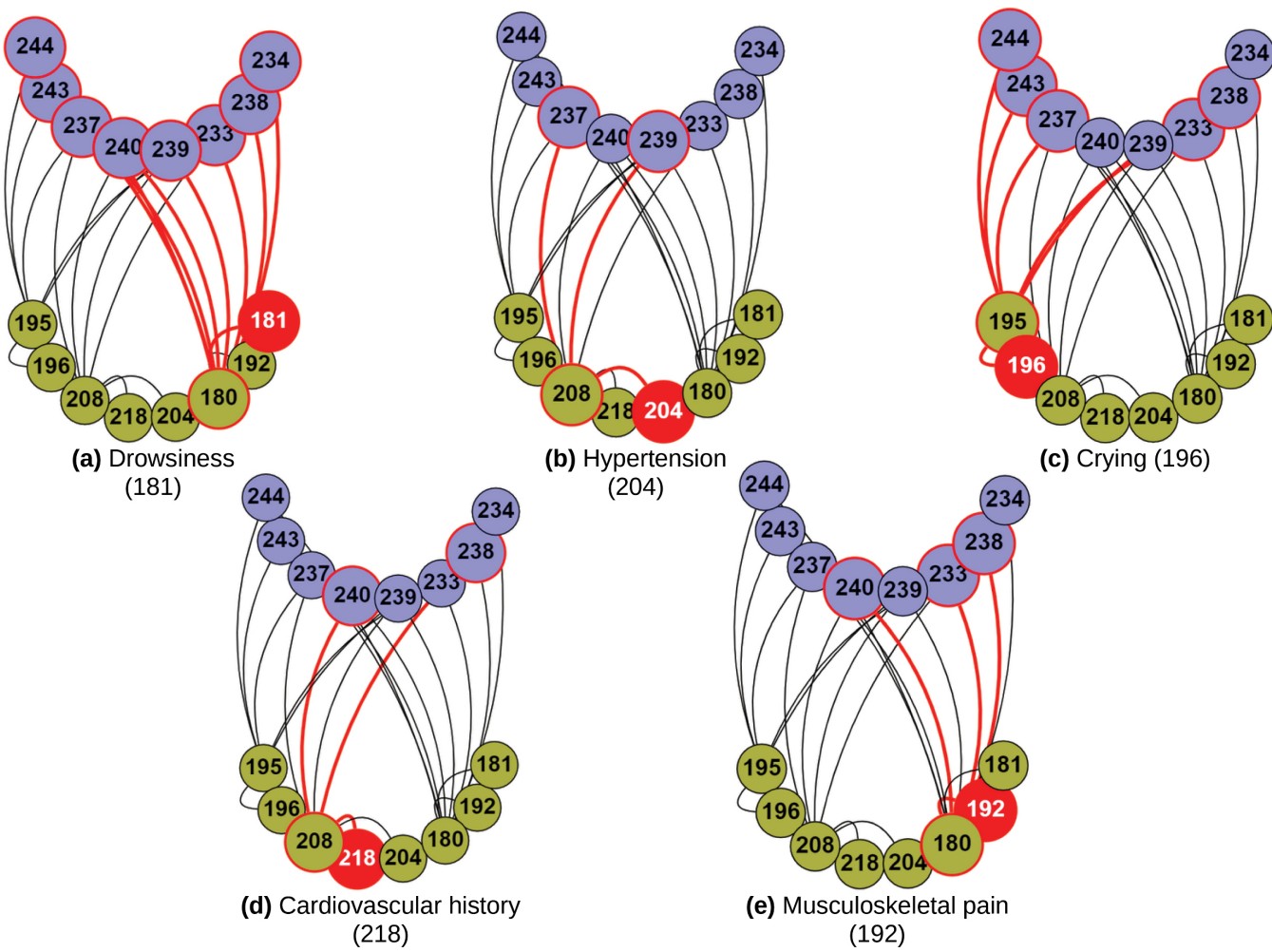

**Fig 14. Selection of the top paths when the source is a risk factor node**. The top 20 shortest paths are presented for each input (risk factors) - output (clinical phenotype) pair. Nodes in the risk factor layer are shown in yellow and those in the phenotype layer in blue. The paths for a particular source out of the five chosen are shown in red: drowsiness, hypertension, crying, cardiovascular history and musculoskeletal pain. For clarity, nodes are labeled numerically in the figure, and the corresponding variable names are provided in S6 Table. For detailed high-resolution paths, please refer to https://dsb-lab.github.io/risk_paths/.

clinical dementia. This aligns with the known progression of AD pathology, in which amyloid accumulation precedes and likely contributes to widespread network dysfunction as the disease evolves.

## Discussion

Our study addresses the complexity of AD from a holistic perspective, exploring the connections between the genetic, molecular, and clinical factors that underlie this pathology. Using a systems biology approach, we have integrated genomics, brain imaging, and clinical data to analyse AD. We use multilayer network analysis and deep phenotyping to unravel the complex mechanisms underlying the disease. Our research reveals significant connections between different biological features and the clinical manifestation of the disease. Our path analysis identified the involvement of FDG PET in most of the key paths supporting the role

of brain hypometabolism in the disease. As such, AD was proposed as a metabolic disease [33–37]. These findings could significantly improve our understanding of AD.

Results obtained in previous studies support our finding. For instance, there is a clear influence of metabolic changes on the disease in the regions of the angular gyrus and posterior cingulate cortex [38]. The relationship of these areas with the cognitive dysfunction associated with AD is probably related to their involvement in various cognitive processes such as attention, visuospatial processing, and memory. This is consistent with the brain hypometabolism observed in Alzheimer's disease patients [39–42]. [43] explores how restoring glucose metabolism in the hippocampus can improve cognitive function. In addition, neuronal dysfunction in these areas may contribute to the manifestation of emotional symptoms, such as depression and mood swings [44,45]. Cardiovascular history has also been associated with an increased risk of cognitive decline and dementia [46], including Alzheimer's disease and FDG hypometabolism in AD-sensitive regions [47,48]. Several observational studies have shown the potential beneficial role of antihypertensive treatment in preventing cognitive decline. However, these associations are complex and not fully elucidated [49].

Interestingly, A$\beta$ and tau levels in the CSF are not very relevant in our global paths: they do not show much direct association with cognition, probably because their effect is reflected much more in PET and MRI nodes and these are the ones that affect phenotype the most, making the effect of molecular nodes very diffuse, as other studies have pointed out [50]. However, given that APP levels in CSF represent the origin of some of the identified global paths, it is interesting to focus prevention on reducing their proliferation. Therapies specifically aimed at modulating the activity of those elements have proved futile, probably because they are carried out when cognitive symptoms are already present and their effect cannot be reversed [51]. On the contrary, these results differ in the analysis by groups. In early stages (control), information flow remains more distributed, and traditional biomarkers may not yet dominate the network structure. As the disease advances (MCI and AD), specific nodes, especially those from the PET layer and molecular layer (A$\beta$42), become increasingly central. This group-level comparison provides further support for the hypothesis that higher-level features such as PET metabolism and symptomatic risk factors mediate and amplify the effect of underlying genetic and molecular variation, shaping the observed phenotype through dynamically structured pathways.

On the other hand, our data suggests that early detection of these biomarkers could allow preventive or therapeutic interventions aimed at modifying disease progression before clinical symptoms appear. Moreover, our paths show that other symptoms such as low energy, depressive mood, crying, and gastrointestinal issues may play a bridging role between physiological disruptions and cognitive decline. This finding highlights the importance of considering these factors not merely as secondary outcomes or quality-of-life markers, but as potentially influential components in the disease process itself. Their centrality in the network supports the rationale for incorporating psychosocial screening and targeted behavioral interventions as part of a broader, multimodal strategy for managing AD progression [52]. It is also noteworthy the recurring presence in the path analysis of the MOCA variable, which may serve as a sensitive indicator of cognitive change that reflects the influence of diverse underlying mechanisms. While caution is warranted in interpreting MOCA as a standalone diagnostic measure, our findings support its utility as a composite cognitive outcome that is highly responsive to multimodal perturbations, making it a valuable target for future studies.

In the past, network techniques have shown to be very successful in offering useful insights into the complicated molecular basis of illnesses, transcending the usual viewpoints centered on single genes and pathways. Within the traditional network framework, molecules are interconnected based on their biological interactions, and by studying the structure and dynamics

of these interaction networks, it becomes possible to uncover disease modules and nonlinear pathways [53]. Weighted gene co-expression networks have been used, for instance, to identify groups of genes (modules) involved in various activated pathways leading to hypertension [54] or to breast cancer and AD [55]. Recently, these approaches have evolved to encompass multiple biological layers: in [56] protein-protein interactions related to essential hypertension were studied through network analyses, also in [57] diverse biological processes such as membrane potential dynamics and signaling were studied within insulin-secreting cells. Approaches based on multilayer networks have also produced notable results in the study of cancer [58–63] and multiple sclerosis [11]. Moreover, new lines of research have been opened up by the use of network-based models, for example by suggesting a possible connection between age-related macular degeneration (nAMD) and neurodegenerative disorders such as AD, schizophrenia and Parkinson's disease [64]. Here we have applied a multilayer network analysis to represent the flow of events that underlies the phenotype of a complex disease such as AD.

Our multilayer network analysis enabled an examination of the interplay between various biological scales in Alzheimer's disease, revealing paths that connect six scales (genomics, molecular, PET, MRI, risk factors, and phenotype) through statistical associations. The analysis provides evidence for information flow across different scales, with the imaging levels (PET and MRI) emerging as particularly informative. The layers interconnect with diverse strengths, and information is modulated as it propagates across them [65,66].

Prior research has sought to establish a direct connection between the genomic layer and phenotypes in various complex diseases, including AD [67,68]. However, genotypes, A$\beta$ and tau deposition, and brain metabolism alone have a limited ability to predict the phenotype [69,70]. Our findings incorporate omics, imaging, and phenotype data, underscore the significance of conceptualizing AD as a multiscale condition. Additionally, the identified paths may serve as potential targets for future personalized medicine treatments in AD.

The data obtained from the ADNI cohort proved to be rich, covering a broad spectrum of scales. However, certain limitations were encountered during the extraction and analysis of variables. While the cohort's overall sample size was sufficient for detecting significant correlations, some specific layers, such as the genomics one, had smaller sample sizes. This limitation has an impact on both the construction of networks and the identification of paths. Additionally, the analysis had to adopt a cross-sectional approach due to inadequate follow-up in many participants. The inclusion of longitudinal data across all six layers would enhance the value of future studies.

## Conclusion

In summary, this study examined the functional connections among various scales of biological data of a complex disease with a complex genetic basis, namely AD. A key finding of this study, observed from the computed principal paths, was the prominent role of cerebral hypometabolism, specifically in the posterior cingulate, as a significant predictor of the average cognitive phenotype. Additionally, combinations of symptomatic variables related to mental health (such as depression, mood swings, and drowsiness) and vascular features (including hypertension and cardiovascular history) were also crucial in explaining the observed cognitive phenotype. The approach to understanding complex biological systems through network science is a very active interdisciplinary research field that is gaining more attention nowadays. Multilayer networks offer several advantages in comparison to traditional network approaches because of their enormous potential to explore the organisation and connections of the different biological layers in both health and disease, making it a promising

tool for future efforts in this area of research. This approach could be applied to other neurodegenerative diseases and autoimmune disorders.

## Supporting information

**S1 Table. Description of the variables in the Genetic dataset.**
(PDF)

**S2 Table. Description of the variables in the Molecular dataset.**
(PDF)

**S3 Table. Description of the variables in the PET dataset.**
(PDF)

**S4 Table. Description of the variables in the MRI dataset.**
(PDF)

**S5 Table. Description of the variables in the Phenotype dataset.**
(PDF)

**S6 Table. Description of the variables in the Risk Factors dataset.**
(PDF)

**S1 Fig. Connectivity matrices between phenotypic variables for each input-specific path analysis.**
(TIF)

**S7 Table. Detailed connectivity of FDG PET nodes with other layers.**
(PDF)

**S8 Table. Top 10 genetic input paths in the control group.**
(PDF)

**S9 Table. Top 10 molecular input paths in the control group.**
(PDF)

**S10 Table. Top 10 PET input paths in the control group.**
(PDF)

**S11 Table. Top 10 MRI input paths in the control group.**
(PDF)

**S12 Table. Top 10 risk factors input paths in the control group.**
(PDF)

**S13 Table. Top 10 genetic input paths in the MCI group.**
(PDF)

**S14 Table. Top 10 molecular input paths in the MCI group.**
(PDF)

**S15 Table. Top 10 PET input paths in the MCI group.**
(PDF)

**S16 Table. Top 10 MRI input paths in the MCI group.**
(PDF)

**S17 Table. Top 10 risk factors input paths in the MCI group.**
(PDF)

**S18 Table. Top 10 genetic input paths in the AD group.**
(PDF)

**S19 Table. Top 10 molecular input paths in the AD group.**
(PDF)

**S20 Table. Top 10 PET input paths in the AD group.**
(PDF)

**S21 Table. Top 10 MRI input paths in the AD group.**
(PDF)

**S22 Table. Top 10 risk factors input paths in the AD group.**
(PDF)

## Acknowledgments

Data collection and sharing for this project was funded by the Alzheimer's Disease Neuroimaging Initiative (ADNI) (National Institutes of Health Grant U01 AG024904) and DOD ADNI (Department of Defense award number W81XWH-12-2-0012). ADNI is funded by the National Institute on Aging, the National Institute of Biomedical Imaging and Bioengineering, and through generous contributions from the following: AbbVie, Alzheimer's Association; Alzheimer's Drug Discovery Foundation; Araclon Biotech; BioClinica, Inc.; Biogen; Bristol-Myers Squibb Company; CereSpir, Inc.; Cogstate; Eisai Inc.; Elan Pharmaceuticals, Inc.; Eli Lilly and Company; EuroImmun; F. Hoffmann-La Roche Ltd and its affiliated company Genentech, Inc.; Fujirebio; GE Healthcare; IXICO Ltd.; Janssen Alzheimer Immunotherapy Research & Development, LLC.; Johnson & Johnson Pharmaceutical Research & Development LLC.; Lumosity; Lundbeck; Merck & Co., Inc.; Meso Scale Diagnostics, LLC.; NeuroRx Research; Neurotrack Technologies; Novartis Pharmaceuticals Corporation; Pfizer Inc.; Piramal Imaging; Servier; Takeda Pharmaceutical Company; and Transition Therapeutics. The Canadian Institutes of Health Research is providing funds to support ADNI clinical sites in Canada. Private sector contributions are facilitated by the Foundation for the National Institutes of Health (www.fnih.org). The grantee organization is the Northern California Institute for Research and Education, and the study is coordinated by the Alzheimer's Therapeutic Research Institute at the University of Southern California. ADNI data are disseminated by the Laboratory for Neuro Imaging at the University of Southern California.

## Author contributions

**Conceptualization:** Juan Domingo Gispert, Jordi Garcia-Ojalvo, Pablo Villoslada.

**Data curation:** Elena Lara-Simon.

**Formal analysis:** Elena Lara-Simon, Pablo Villoslada.

**Investigation:** Elena Lara-Simon, Jordi Garcia-Ojalvo, Pablo Villoslada.

**Methodology:** Pablo Villoslada.

**Project administration:** Pablo Villoslada.

**Resources:** Pablo Villoslada.

**Software:** Elena Lara-Simon.

**Supervision:** Juan Domingo Gispert, Jordi Garcia-Ojalvo, Pablo Villoslada.

**Validation:** Jordi Garcia-Ojalvo, Pablo Villoslada.

**Writing – original draft:** Elena Lara-Simon.

**Writing – review & editing:** Juan Domingo Gispert, Jordi Garcia-Ojalvo, Pablo Villoslada.

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
