## [Decision Letter · Decision Letter 0]

9 May 2025

PCOMPBIOL-D-24-01994

Multiscale networks in Alzheimer’s disease identify brain hypometabolism as central across biological scales

PLOS Computational Biology

Dear Dr. Villoslada,

Thank you for submitting your manuscript to PLOS Computational Biology. After careful consideration, we feel that it has merit but does not fully meet PLOS Computational Biology's publication criteria as it currently stands. Therefore, we invite you to submit a revised version of the manuscript that addresses the points raised during the review process.

Please submit your revised manuscript within 90 days. If you will need more time than this to complete your revisions, please reply to this message or contact the journal office at ploscompbiol@plos.org. Please include the following items when submitting your revised manuscript:

We look forward to receiving your revised manuscript.

Kind regards,

Feixiong Cheng, Ph.D.

Academic Editor

PLOS Computational Biology

Pedro Mendes

Section Editor

PLOS Computational Biology

**Additional Editor Comments:**

Additionally to addressing the reviewer comments, the code for this work does not seem to be available (only the networks). Please provide access to the code according to the PLOS Computational Biology policies (https://journals.plos.org/ploscompbiol/s/code-availability)

**Journal Requirements:**

At this stage, the following Authors/Authors require contributions: Elena Lara-Simon, Juan Domingo Gispert, Jordi Garcia-Ojalvo, and Pablo Villoslada. Please ensure that the full contributions of each author are acknowledged in the "Add/Edit/Remove Authors" section of our submission form.

Potential Copyright Issues:

- Figure 5. Please confirm whether you drew the images / clip-art within the figure panels by hand. If you did not draw the images, please provide (a) a link to the source of the images or icons and their license / terms of use; or (b) written permission from the copyright holder to publish the images or icons under our CC BY 4.0 license. Alternatively, you may replace the images with open source alternatives. See these open source resources you may use to replace images / clip-art:

5) Please amend your detailed Financial Disclosure statement. This is published with the article. It must therefore be completed in full sentences and contain the exact wording you wish to be published. Please ensure that the funders and grant numbers match between the Financial Disclosure field and the Funding Information tab in your submission form. Note that the funders must be provided in the same order in both places as well.

**Reviewers' comments:**

Reviewer's Responses to Questions

**Comments to the Authors:**

Reviewer #1: Review Uploaded As Attachment

Reviewer #2: In the manuscript, Lara-Simon et al. presents a systems biology study applying multilayer network analysis to integrate multi-layered AD data (genomics, CSF/plasma biomarkers, PET/MRI imaging, clinical phenotypes, and risk factors) to investigate the complex interplay across different biological scales. The authors construct mutual-information-based network for each layer and use the dynamical Boolean simulations to identify potential paths that link molecular changes to clinical phenotype for AD. Via network and path analysis, they conclude that FDG PET hypometabolism may be as a central hub connecting different layers in AD phenotypes (including cognitive impairment and risk factors). This is an interesting study that integrate microscopic and macroscopic layers to address the AD complexity. However, I have some concerns for both methodology and results.

Major issues:

1. The title emphasizes "brain hypometabolism," but this key finding is not mentioned in the Abstract. In addition, in the entire Results section, only lines 305-307 claimed MRI or PET connections associated with metabolic processes. The authors primarily claim hypometabolism findings in the Discussion section. Is there additional analysis that supports this primary conclusion? Given that MRI or PET provided most of the variables used for network construction (SI sheets), do the network methods justify the variable count across different layers?

2. Since the multilayer network approach has already been explored in a previous study (reference 11), the authors should state what methodological novelty in this manuscript.

3. The study aims to address heterogeneity but primarily focuses on paths explaining the phenotype. The manuscript only highlights the network changes in AD, it would better to include MCI and Control as well and compare the differences from network and path perspectives.

4.In the path analysis, some phenotypes, such as MOCA and 5 risk factors (HMSCORE, AXDIARRH, AXENERGY, AXDPMOOD, AXCRYING) will be linked regardless of the start point is. How to explain this result?

5. In the path analysis, one phenotype may link to multiple other phenotypes, could the authors add a SI table to show these results? It may be helpful to understand the AD complexity.

Minor issues:

1. Line 354: “To identify the causal logic backbone of the network…” It may be too strong because the Boolean simulations might not reflect causal cascades.

2. Do all samples from collected ADNI cohort have all variables in each layer? If not, how the author constructs the network by considering all variables, e.g., by computing inter-layer MI? These details should be clarified.

3. Inconsistent count of subjects: the methods say 1776 participants (888 MCI, 348 AD, 540 controls), but the Results mention 622 controls, 807 MCI, 533 AD.

4. Some inaccurate claims:

line 79: “more than 30 genetic variants”,

abstract: “average cognitive phenotype”,

line 426, 546: should include p value for “significantly”

5. Sometimes “subjects with mild AD” are mentioned, elsewhere just “AD patients”. They are different, please keep consistent.

6. Fig 9-13: missing color bars.

**Have the authors made all data and (if applicable) computational code underlying the findings in their manuscript fully available?**

Reviewer #1: **No: **I could see the GitHub link with the networks generated for the paper. However, I was unable to find the relevant code. If it has been uploaded, an easier access to it would be better.

Reviewer #2: **No: **Code is currently not accessible.

PLOS authors have the option to publish the peer review history of their article (what does this mean?). If published, this will include your full peer review and any attached files.

Reviewer #1: No

Reviewer #2: No

**Figure resubmission:**
---

## [Decision Letter · Decision Letter 1]

22 Aug 2025

PCOMPBIOL-D-24-01994R1

Multiscale networks in Alzheimer’s disease identify brain hypometabolism as central across biological scales

PLOS Computational Biology

Dear Dr. Villoslada,

Thank you for submitting your manuscript to PLOS Computational Biology. After careful consideration, we feel that it has merit but does not fully meet PLOS Computational Biology's publication criteria as it currently stands. Therefore, we invite you to submit a revised version of the manuscript that addresses the points raised during the review process.

Please submit your revised manuscript within 30 days. If you will need more time than this to complete your revisions, please reply to this message or contact the journal office at ploscompbiol@plos.org. Please include the following items when submitting your revised manuscript:

We look forward to receiving your revised manuscript.

Kind regards,

Feixiong Cheng, Ph.D.

Academic Editor

PLOS Computational Biology

Pedro Mendes

Section Editor

PLOS Computational Biology

**Additional Editor Comments:**

The authors are suggested to address additional critiques from the reviewer.

**Journal Requirements:**

Please ensure that the funders and grant numbers match between the Financial Disclosure field and the Funding Information tab in your submission form. Note that the funders must be provided in the same order in both places as well.

State what role the funders took in the study. If the funders had no role in your study, please state: "The funders had no role in study design, data collection and analysis, decision to publish, or preparation of the manuscript.".

**Reviewers' comments:**

Reviewer's Responses to Questions

**Comments to the Authors:**

Reviewer #2: The authors well addressed my concerns. Overall, the manuscript looks much improved both in clarity and content. There are some minor comments:

1. As FDG PET was found as a key path for AD, but it is still not clear how this node connects with other nodes in other layers. Particularly for FDG PET, please describe more about its upstream and downstream connections to nodes in other layers. A small schematic table/figure would help readers.

2. The names of “control group” in the text or tables are not consistent: the names “normal control, health control or Control” should be standardized.

3. Page 11 line 303: “A threshold was implemented to examine…” Please clarify the threshold explicitly.

4. Please improve Figures 6-14 to make the labels much clearly to readers.

5. Glad to see the codes have been uploaded. For transparency and reproducibility, please deposit de-identified source data in github as well.

**Have the authors made all data and (if applicable) computational code underlying the findings in their manuscript fully available?**

Reviewer #2: **No: **Source data for the codes are not shown.

PLOS authors have the option to publish the peer review history of their article (what does this mean?). If published, this will include your full peer review and any attached files.

Reviewer #2: No

**Figure resubmission:**
---

## [Editor Report · Decision Letter 2]

2 Oct 2025

Dear Dr. Pablo Villoslada,

We are pleased to inform you that your manuscript 'Multiscale networks in Alzheimer’s disease identify brain hypometabolism as central across biological scales' has been provisionally accepted for publication in PLOS Computational Biology.

Best regards,

Feixiong Cheng, Ph.D.

Academic Editor

PLOS Computational Biology

Pedro Mendes

Section Editor

PLOS Computational Biology

The authors have addressed concerns from previous reviewers.

---

## [Editor Report · Acceptance letter]

PCOMPBIOL-D-24-01994R2

Multiscale networks in Alzheimer’s disease identify brain hypometabolism as central across biological scales

Dear Dr Villoslada,

I am pleased to inform you that your manuscript has been formally accepted for publication in PLOS Computational Biology. Your manuscript is now with our production department and you will be notified of the publication date in due course.

With kind regards,

Lilla Horvath
